



# Model development in practice: A comprehensive update to the boundary layer schemes in HARMONIE-AROME cycle 40

Wim C. de Rooy[1], Pier Siebesma[1,2], Peter Baas[2], Geert Lenderink[1], Stephan de Roode[2], Hylke de Vries[1], Erik van Meijgaard[1], Jan Fokke Meirink[1], Sander Tijm[1], and Bram van 't Veen[1]

[1]Research & Development Weather and Climate models, Royal Netherlands Meteorological Institute, De Bilt, Utrecht, PO Box 201 3730AE, The Netherlands
[2]Department of Geoscience & Remote Sensing, Delft University of Technology, Delft , Stevinweg 1, 2628CN, The Netherlands

**Correspondence:** Wim de Rooy (rooyde@knmi.nl)

**Abstract.** The parameterised description of subgrid-scale processes in the clear and cloudy boundary layer has a strong impact on the performance skill in any Numerical Weather Prediction (NWP) or climate model and is still a prime source of uncertainty. Yet, improvement of this parameterised description is hard because operational models are highly optimised and contain numerous compensating errors. Therefore, improvement of a single parameterised aspect of the boundary layer often results in

an overall deterioration of the model as a whole. In this paper we will describe a comprehensive integral revision of three parameterisation schemes in the HARMONIE-AROME model that together parameterise the boundary layer processes: the cloud scheme, the turbulence scheme, and the shallow cumulus convection scheme. One of the major motivations for this revision is the poor representation of low clouds in the current model cycle. The new revised parametric descriptions provide not only an improved prediction of low clouds but also of precipitation. Both improvements can be related to a stronger accumulation

of moisture under the atmospheric inversion. The three improved parameterisation schemes are included in a recent update of the HARMONIE-AROME configuration, but its description and the insights in the underlying physical processes are of more general interest as the schemes are based on commonly applied frameworks. Moreover, this work offers an interesting look behind the scenes of how parameterisation development requires an integral approach and a delicate balance between physical realism and pragmatism.

## 1  Introduction

Owing to ever growing computer resources, numerical resolution of weather and climate models steadily refines. Presently, limited area models operate routinely at resolutions of around $1\,\mathrm{km}$ and the first global intercomparison project for global storm-resolving models at resolutions of $5\,\mathrm{km}$ demonstrates that deep convective overturning processes are at least partly resolved by the new generation of weather and climate models (Stevens et al., 2019).





Prime atmospheric processes that remain to be parameterised at these scales are turbulent transport in the boundary layer, shallow cumulus convection, radiation, and cloud micro- and macrophysical processes of unresolved clouds. Traditionally, parameterisation of these processes have been developed as independent building blocks. The turbulence scheme describes the transport of heat, moisture and momentum by the small-scale turbulent eddies in the boundary layer, whereas the convection scheme represents the transport by the more larger-scale organised convective plumes. The cloud scheme aims to estimate the cloud fraction and the amount of condensed water.

Nowadays it is recognized that the latter three parameterisation schemes need to be tightly coupled as illustrated in Figure 1. The cloud scheme requires information on the subgrid-scale variability of moisture and temperature as produced by the turbulence and convection scheme. Vice versa, the mixing by turbulence in the cloud boundary layer depends strongly on the cloud fraction. Clearly, optimisation of only one scheme will likely deteriorate the performance of another coupled scheme. This is why we describe in this paper the revision and optimisation of a tightly coupled triplet of parameterisation schemes for boundary layer turbulence, shallow cumulus convection and clouds.

As stated by Jakob (2010): "Whereas early parameterisations development was aimed at finding suitable simple statistical relationships, modern parameterisations constitute complex conceptual models of the physical processes they are aiming to represent". Indeed, more physically based parameterisations should be preferred as long as they improve the representation of essential processes, i.e. processes that significantly influence the resolved-scale variables. On the other hand, extra complexity in parameterisations should only be added, if this does not imply introducing extra tunable parameters that can not be constrained. Finding an acceptable level of physical realism and complexity without introducing too many tunable parameters that could give rise to over-fitting, or even lead to an unstable system, is an important theme in this study.

The here investigated parameterisations are part of the convection-permitting HARMONIE-AROME numerical weather prediction (Bengtsson et al., 2017) and climate model (Belušić et al., 2020). Bengtsson et al. (2017), from hereon B17, present the HARMONIE-AROME configuration of cycle 40 (cy40) including a brief description of the reference model physics, noted as cy40REF. In contrast to B17, this paper provides a comprehensive description of the cloud, turbulence and cloud scheme. Moreover, we present numerous adjustments and improvements to the reference set-up, included in a version referred to as cy40NEW. All these adjustments are accepted as the default options in the next release of HARMONIE-AROME, cycle 43.

The primary goal of these adjustments is to improve on what is considered as one of the most important model deficiencies of HARMONIE-AROME cy40: a substantial underestimation of low cloud amount and overestimation of cloud base height.

The presented changes in the parameterisation schemes are primarily based on process studies and theoretical considerations. For example, long-term Single Column Model (SCM) runs are used to evaluate the turbulence scheme in terms of theoretical flux-gradient relationships, following the procedure of Baas et al. (2017). Based on these results important modi-





fications are made to the turbulence scheme. Additionally, several model intercomparison studies covering shallow cumulus, stratocumulus and dry stable boundary-layer conditions are used, most of which were based on observations collected during field campaigns. For these intercomparison cases, results of the Dutch Large Eddy Simulation (DALES (Heus et al., 2010)) are compared in detail with SCM runs of HARMONIE-AROME. Finally, for the optimisation of the remaining uncertain parameters, we follow a more pragmatic approach by utilising 3D model runs.

This paper can be considered as a description of a substantial model update concerning several parameterisation schemes. Although the parameterisations are embedded in the HARMONIE-AROME model, we believe that our findings are more generally applicable in NWP and Climate models. Even though the schemes in other models may differ in details, the parameterisations in HARMONIE-AROME are based on widely applied frameworks: a statistical cloud scheme, a (bulk) mass flux convection scheme and a Turbulence Kinetic Energy (TKE) based turbulence scheme. Hence, the here described modifications and the impact of certain parameters, or combinations of them, are useful for any atmospheric model that requires a parameterised representation of the clear and cloudy boundary layer.

We start with a description of the convection, turbulence and cloud scheme in section 2. Section 2.2 provides the first complete and detailed description of the shallow convection scheme. Documentation of the turbulence scheme can be found in Lenderink and Holtslag (2004) and Bengtsson et al. (2017). Therefore, only the parameters involved in the adjustments to the turbulence scheme are introduced in section 2.3. Because of the comprehensive update to the statistical cloud scheme, a full description is provided in section 2.4. Some of the adjustments introduced in section 2 might seem arbitrary at first sight. However, section 3 describes the experiments to motivate these adjustments. Several modifications are based on a comparison of SCM runs with LES for the idealised case ARM (Section 3.1). SCM runs are also used to optimise the turbulence scheme against theoretical flux gradient relationships in section 3.2. Section 3 further demonstrates the substantial improvements with the new configuration. For this, idealised cases of stratocumulus (section 3.3), shallow convection (section 3.1.2) and moderately stable conditions (section 3.2) are used, as well as full 3D model runs in section 3.4. Finally in section 4 the discussions and conclusions are presented.

## 2 Parameterisation schemes

### 2.1 General Framework

Before giving a more detailed description of the involved parameterisations in the next sections, we start with introducing the general parameterisation framework of the clear and cloud topped boundary layer. The grid-box averaged prognostic equations



for the liquid water potential temperature $\theta_\ell$ and the total water specific humidity $q_t$ can be written as

$$\overline{\mathrm{D}}_t\overline{\theta}_\ell = -\frac{1}{\rho}\frac{\partial\rho\overline{w'\theta'_\ell}}{\partial z} + Q_{\mathrm{rad}} \tag{1a}$$

$$\overline{\mathrm{D}}_t\overline{q}_{\mathrm{t}} = -\frac{1}{\rho}\frac{\partial\rho\overline{w'q'_{\mathrm{t}}}}{\partial z} - G \tag{1b}$$

where $\rho$ is the average density, $w$ the vertical velocity, $G$ the autoconversion rate from condensed cloud water to rain water and $Q_{\mathrm{rad}}$ the radiative heating tendency. The primes denote deviation from the grid mean values. The operator $\mathrm{D}_t$ represent a total time derivative while the overbars denote the grid box mean for an arbitrary variable $\phi$. Note that the condensation and

evaporation tendencies are not present because we use a formulation in terms of moist conserved variables. The terms on the right hand side of (1) are all subgrid-scale and require a parameterised description.

The turbulent fluxes are parameterised using the Eddy-Diffusivity Mass-Flux (EDMF) framework (Siebesma and Teixeira, 2000). This framework has been designed in order to facilitate a unified description of the turbulent transport in the dry convective boundary layer (Siebesma et al., 2007) and the cloud topped boundary (Soares et al., 2004; Rio and Hourdin,

2008). More recent refinements and developemnts can be found in Neggers et al. (2009); Sušelj et al. (2013). The EDMF approach is inspired on the notion that cumulus convection is usually rooted in the subcloud layer from which rising thermals transport moist buoyant air into the cumulus clouds aloft. It is therefore natural to decompose the turbulence into organised convective updrafts and a remaining part consisting of smaller-scale turbulent eddies

$$\overline{w'\phi'} = \overline{w'\phi'}^{\mathrm{turb}} + \overline{w'\phi'}^{\mathrm{conv}}. \tag{2}$$

As long as the updraft fraction $a_{\mathrm{u}}$ is much smaller than unity, the convective transport can be conveniently parameterised in a mass flux (MF) framework as

$$\overline{w'\phi'}^{\mathrm{conv}} \approx \frac{\mathcal{M}_{\mathrm{u}}}{\rho}\left(\phi_u - \overline{\phi}\right) \quad , \quad \mathcal{M}_{\mathrm{u}} = \rho a_{\mathrm{u}}w_{\mathrm{u}} \tag{3}$$

where a bulk convective mass flux $\mathcal{M}_{\mathrm{u}}$ has been introduced and where $w_{\mathrm{u}}$ denotes the vertical velocity in the updraft. The remaining small-scale local turbulence is approximated by vertical diffusion by means of an eddy diffusivity (ED) approach

$$\overline{w'\phi'}^{\mathrm{turb}} \approx -K\frac{\partial\overline{\phi}}{\partial z} \tag{4}$$

which completes the EDMF framework in its simplest form. Note that the parameterisation task is now reduced to finding appropriate expressions for the mass flux $\mathcal{M}_{\mathrm{u}}$ the updraft fields $\phi_{\mathrm{u}}$ and the eddy diffusivity $K$. One prime advantage of the EDMF approach is that the mass flux description of the updrafts can be active for both the clear and cloud topped boundary layer so that the transition between these regimes can occur in a more continuous manner without the need of explicit switches

or trigger functions.

There is a strong interplay between turbulence and convection (see Fig. 1). For example the transport of heat by the convective thermals produced by the mass flux scheme will establish a neutral to slightly stable stratification in the upper



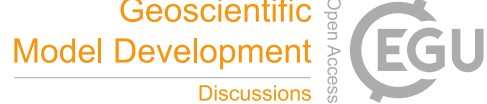

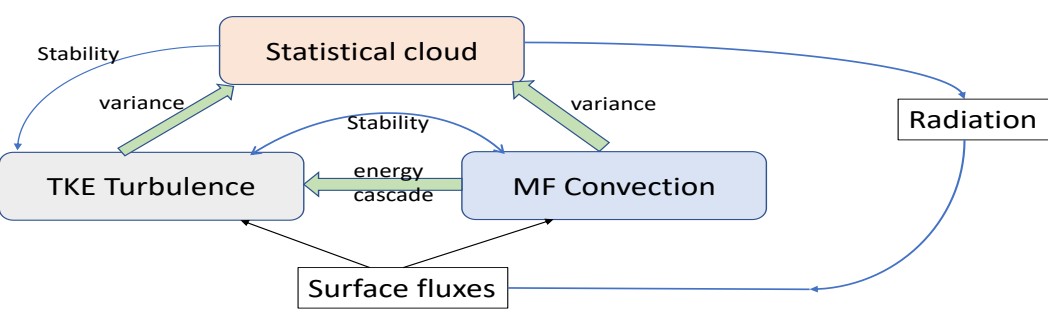

**Figure 1.** Schematic diagram illustrating the direct (thick arrows) and indirect (thin arrow) dependencies of parameterisation schemes with a focus on the schemes involved in the modifications.

part of the convective boundary layer, thereby suppressing the diffusive transport by the TKE scheme in this area (Lenderink et al., 2004). Besides, there is also a direct (coded) link between these schemes as the mass flux is used as a source term in a

TKE budget equation that is used to parameterise the eddy diffusivity $K$ (see Fig. 1). This interaction mimicks the turbulence energy cascade in which turbulent kinetic energy cascades from the larger eddies down to the smaller eddies and will be further discussed in sections 2.3 and 3.1.1.

The last parameterisation involved in the modifications is the cloud scheme. The task of the cloud scheme is to estimate the subgrid-scale cloud fraction and the condensed water. A common approach to calculate cloud cover and condensed

water is to assume a subgrid-scale distribution of humidity and temperature and to determine the cloud cover as the fraction of the distribution above saturation. A key element in such a statistical cloud scheme is the estimate of the subgrid-scale variance of the relative humidity. Important contributions to this variance are the convective (3) and turbulent (4) transport, establishing a strong link between the cloud scheme and the turbulence and convection parameterisations (Fig. 1).

The specific parameterisation implementations in HARMONIE-AROME, are descibed in more detail in the upcom-

ing subsections. The parameterisations of the convective mass flux $\mathcal{M}_u$ and the updraft fields $\phi_u$ are discussed in subsection 2.2. The eddy diffusivity parameterisation is discussed in subsection 2.3 and the cloud scheme finally in subsection 2.4.





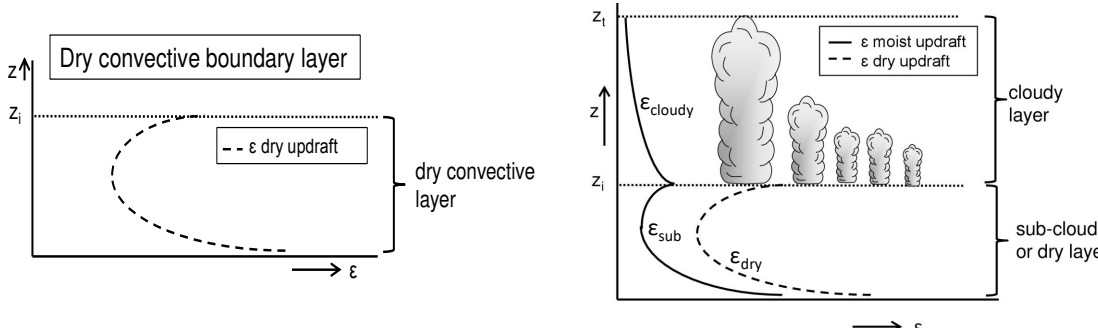

**Figure 2.** Schematic diagrams of the convective boundary layer regimes and the corresponding entrainment formulations (8) of the dry (dashed line) and moist (solid line) updraft. The inversion height and cloud top height are respectively denoted as $z_i$ and $z_t$. Note that $z_i$ can be different for the moist and dry updraft and is therefore referred to as $z_{i,dry}$ and $z_{lcl}$ respectively. The shape of the entrainment profiles reflects the inverse dependency on the vertical velocity of the updraft (section 2.2.1). This is a modified figure of Fig. 4 in B17

## 2.2 Shallow convection scheme

The mass flux description is based on a dual mass flux approach (see e.g. Neggers et al. (2009), from hereon N09) in which instead of one bulk updraft as in Eq. (3), we distinguish two updrafts: 1) a dry updraft describing all the thermals that do not convert into saturated updrafts in the cloud layer and 2) a moist updraft representing all updrafts that do reach the lifting condensation level (lcl) and continue their ascent in the cloud layer.

$$\rho \overline{w'\phi'}^{\mathrm{conv}} \approx \mathcal{M}_{\mathrm{dry}}\left(\phi_{\mathrm{u,dry}} - \overline{\phi}\right) + \mathcal{M}_{\mathrm{moist}}(\phi_{\mathrm{u,moist}} - \overline{\phi}) \tag{5}$$

As illustrated schematically in Fig. 2 we distinguish two different convective boundary layer regimes; dry convective boundary layers with only a dry updraft and cloud topped boundary layers with a dry and a moist updraft. Note that in contrast with B17 and N09, a stratocumulus topped boundary layer in cy40NEW still uses a dry updraft (further discussed in Section 3.3).

The updraft profiles $\phi_{\mathrm{u,i}}$ of updraft i (i $\in \{\mathrm{dry, moist}\}$) are determined by a conventional entraining plume model

$$\frac{\partial \phi_{\mathrm{u,i}}}{\partial z} = -\varepsilon_{\mathrm{k}}(\phi_{\mathrm{u,i}} - \overline{\phi}) + \mu_\phi \tag{6}$$

where $\epsilon_{\mathrm{k}}$ denotes the fractional entrainment rate of the updraft and where $\mu_\phi$ represents cloud microphysical effects such as precipitation generation in the updraft (parameterised according to N09). The subscript k refers to different entrainment formulations for the dry updraft, the moist updraft in the subcloud layer and the moist updraft in the cloud layer, i.e. k $\in \{\mathrm{dry, sub, cloudy}\}$. The various entrainment formulations are presented in Section (2.2.1).

The updrafts are initialised at the lowest model level with a temperature and humidity that exceed the mean values at that level. The excess values are determined by assuming that the temperature and humidity are Gaussian distributed with





| PBL regime | updraft fractions |
|---|---|
| stable | $a_{dry} = a_{moist} = 0$ |
| dry convective | $a_{dry} = 0.1 \quad a_{moist} = 0$ |
| shallow convection or strato-cumulus | $a_{moist} = 0.03 \quad a_{dry} = 0.1 - a_{moist}$ |

**Table 1.** Updraft area fractions per PBL regime in cy40NEW

| | dry or sub-cloud | cloudy |
|---|---|---|
| $a$ | $\frac{10}{7}$ | $\frac{2}{3}$ |
| $b$ | $\frac{5}{7}$ | 1 |

**Table 2.** Applied $a$ and $b$ coefficients in the vertical velocity equation (7)

a variance estimated from the surface fluxes. The initalisation temperature and humidity values are then given by their $1 - a_u$

percentiles, where $a_u$ denotes the fractional updraft area. Hence, larger variances and smaller area fractions give stronger excess values. We refer to NO9 for a more detailed description of the updraft initialisation. The updraft area fractions $a_u$ are simply prescribed as fixed fractions as in (B17) instead of the more flexible updraft fractions in N09. These fixed updraft fractions depend on the diagnosed boundary layer regime (Table 1). Like in N09, the total updraft fraction under convective conditions is always $0.1$. How the PBL regime is diagnosed is described in the next Section.


In addition to the updraft model for heat and moisture, a similar updraft equation is used for the vertical velocity $w_u$ that can be used to estimate how deep the the updrafts can penetrate ( i.e. the height where $w_u$ vanishes).

$$\frac{1}{2}\frac{\partial w_{u,i}^2}{\partial z} = a_k B_{u,i} - b_k \varepsilon_k w_{u,i}^2 \qquad \text{with} \qquad B_{u,i} = \frac{g}{\theta_v}(\theta_{v,u,i} - \overline{\theta}_v) \qquad (7)$$

Where $w_{u,i}$, $B_{u,i}$ and $\theta_{v,u,i}$ are resp. updraft vertical velocity, buoyancy, and virtual potential temperature of updraft i. $g$ is the

acceleration of gravity. In Eq. (7) $b_k$ and $a_k$ are constants for dry ($k = \text{dry,sub}$ i.e. dry CBL or subcloud layer) and cloudy ($k = \text{cloudy}$) parts of the boundary layer. Note that Eq. (7) is a highly parameterised vertical velocity equation as effects of pressure are absorbed in the constants $b_k$ and $a_k$ (see e.g. de Roode et al. (2012)).

In the literature, a large variety for values of $a$ and $b$ can be found. Based on LES, de Roode et al. (2012) showed that the

accuracy of the vertical velocity equation in the cloud layer depends on a correct combination of $a$ and $b$. They found good correspondence with LES results for the combination of constants in Bechtold et al. (2001), de Rooy and Siebesma (2010) and





Rio et al. (2010) and we adopt these for the cloud layer (see Table 2) . For dry updraft and sub-cloud layer part of the moist updraft we adopt the formulation of Siebesma et al. (2007) (Table 2).

Fractional entrainment is not only applied in determining the updraft dilution in Eq. (6), but also plays a role in the change of the mass flux with height, according to the following simple budget equation:

$$\frac{\partial \mathcal{M}_\mathrm{u}}{\partial z} = (\varepsilon - \delta)\mathcal{M}_\mathrm{u} \tag{8}$$

where $\delta$, the fractional detrainment, describes the outflow of updraft air into the environment. An accurate description of the lateral mixing between the updraft and the environment is key to every mass flux scheme (see e.g. de Rooy et al. (2013)). Hence, $\varepsilon$ and $\delta$ are described in detail in the next sections.

### 2.2.1    Fractional entrainment

Previously, the entrainment coefficients of the HARMONIE-AROME convection scheme have been discussed only briefly (B17). Here they are described in detail. Further motivation for the parameter settings and adjustments are provided in Section 3.

      We need to specify the fractional entrainment factors, $\varepsilon$, for both updraft types. Moreover, for the moist updraft a

distinction is made between the dry sub-cloud layer and the moist cloudy layer (Fig. 2). As demonstrated by de Rooy and Siebesma (2008) and de Rooy and Siebesma (2010), the fractional entrainment in the cloudy layer is mainly a function of the vertical extent of the cloud layer and reflects the general notion that a deeper cloud layers hosts larger clouds with lower fractional entrainment rates.

      The entrainment formulations for the non-cloudy layers are based on existing, LES-based formulations with the

inversion height, $z_\mathrm{i}$, as a parameter (Siebesma et al., 2007). However, the inversion height is not known a-priori. To provide a first estimate of the inversion height we therefore release a test parcel with an entrainment formulation inversely proportional to the vertical updraft velocity (Neggers et al. (2002) and N09 Eq. 19). The test parcel is only used for diagnostic purposes and does not affect the ultimate convective transport. Also note that here inversion height is actually the height where the dry updraft vertical velocity becomes $0$ (so including the overshoot into the inversion) or the lifting condensation level of the moist

updraft.

      Apart from estimating $z_\mathrm{i}$, the test parcel is also used to provide a first estimate of the boundary layer type. If the updraft does not reach the lifting condensation level the boundary layer type is dry convective with only a dry updraft (left panel Fig. 2). If, on the other hand the test parcel becomes saturated during its rise and condensation takes place, a cloudy boundary layer is diagnosed (right panel Fig. 2). In this case a dry and a moist updraft are considered. In case of a cloudy PBL,

the cloud layer depth is also diagnosed and if it exceeds a threshold (currently set to $4000m$), the model is supposed to resolve





moist convection and only dry convection remains parameterised. Note that this threshold value should decrease with increased spatial resolution.

After determination of the PBL regime and the inversion height with the test updraft, the updraft rise is again calculated but this time with the area fractions from Table 1, leading to different initial excess values, and with the refined
entrainment rates as defined below. Hereby, PBL regime and inversion height may alter but this iteration process converges very rapidly.

*Entrainment of the dry updraft*

For any convective PBL regime we need an entrainment formulation for the dry updraft. Based on LES results for a dry CBL, Siebesma et al. (2007) propose a formulation of $\varepsilon$ as a fixed function of height and we roughly adopt their formulation for the
dry updraft:

$$\varepsilon_{\mathrm{dry}} = c_{\mathrm{dry}} \left( \frac{1}{z + a_1} + \frac{1}{z_{\mathrm{i,dry}} - z + a_2} \right) \tag{9}$$

where $c_{\mathrm{dry}} = 0.4$ (Siebesma et al., 2007) and $z_{\mathrm{i,dry}}$ is the dry updraft inversion height. The shape of $\varepsilon_{dry}$ using Eq. (9) (see Fig. (2)a) reflects the increase in vertical velocity up to the middle of the dry convective boundary layer, resulting in decreasing $\varepsilon$ values. From there the updraft slows down resulting in an increase of $\varepsilon$ until the updraft finally stops at inversion height and
$\varepsilon$ becomes infinitely large. In practice this ill definition of $\varepsilon_{dry}$ is prevented by coefficient $a_2$ (similar to Soares et al. (2004)). Again similar to Soares et al. (2004), $a_1$ is introduced in cy40NEW to prevent very high entrainment values near the surface (see Section 3.1.2) and to reduce the dependence on the height of the lowest model level.

*Entrainment of the moist updraft in the sub-cloud layer*

Also for the entrainment of the moist updraft in the sub-cloud layer (10) we build on the formulation of Siebesma et al. (2007)
(9) and Soares et al. (2004), where the latter uses a similar entrainment formulation as (10) but in a single updraft framework.

$$\varepsilon_{\mathrm{sub}} = c_{\mathrm{moist,sub}} \left( \frac{1}{z + a_1} + \frac{1}{z_{\mathrm{lcl}} - z + \frac{z_{\mathrm{lcl}}}{\frac{\varepsilon_{\mathrm{lcl}}}{c_{\mathrm{moist,sub}}} z_{\mathrm{lcl}} - 1}} \right), \quad for \quad z < z_{\mathrm{lcl}} \tag{10}$$

Formulation (9) for the dry updraft needs to be adapted for the sub-cloud moist updraft for two reasons. Firstly, in contrast with the dry updraft, the moist updraft does not stop at inversion height (or cloud base) and therefore $\varepsilon$ does not approach infinity. Instead, the entrainment at cloud base, noted as $\varepsilon_{z_{\mathrm{lcl}}}$ is set to $0.002 m^{-1}$, a reasonable value according to LES results ((de Rooy
et al., 2013), (Siebesma et al., 2003)). The apparently complicated last term in the dominator of Eq. (10) just ensures that the entrainment approaches its cloud base value apart from the term $a_1$. However, $a_1$ is negligible compared to typical $z_{\mathrm{lcl}}$ values. Secondly, the moist updraft represents stronger thermals than the dry updraft. LES results in Siebesma et al. (2007) reveal that the entrainment of stronger dry thermals (selected by changing the sampling criteria) correspond to smaller $c_{\mathrm{dry}}$ values. Extending this to even stronger thermals that manage to become cumulus clouds, we set $c_{\mathrm{moist,sub}} = 0.2$.





As argued in Appendix B, $\varepsilon_{z_{lcl}} = 0.002 m^{-1}$ replaces $\varepsilon_{z_{lcl}} = \frac{1.65}{z_{lcl}}$ in cy40REF where the dependence on $z_{lcl}$ was included to reflect that deeper boundary layers will contain larger updrafts with relatively small entrainment values.

Similar to Eq. (9), Eq. (10) shows an inverse correlation between updraft vertical velocity and entrainment magnitude (see Fig. 2). Like in Eq. (9), $a_1$ is introduced in Eq. (10) of cy40NEW (see Appendix B and Section 3.1.2).

*Entrainment of the moist updraft in the cloudy layer*

The final entrainment profile to be defined is $\varepsilon_{cloudy}$. In contrast to (Soares et al., 2004) the formulations of $\varepsilon_{sub}$ and $\varepsilon_{cloudy}$ are connected at cloud base. From cloud base, $\varepsilon_{cloudy}$ will normally decrease with height related to increasing vertical velocity. Moreover, our bulk scheme should represent an ensemble of clouds and at higher levels only the largest, and fastest rising thermals, with relatively small entrainment values, will survive. Although the exact shape of LES diagnosed entrainment profiles in the cloud layer will depend on the precise sampling method, a decrease proportional to $z^{-1}$ provides an acceptable 
fit and is used as a parameterisation.

$$\varepsilon_{cloudy} = \frac{1}{z - z_{lcl} + \frac{1}{\varepsilon_{z_{lcl}}}}, \quad for \quad z_{lcl} \leq z \tag{11}$$

With, as mentioned before, $\varepsilon_{z_{lcl}} = 0.002 m^{-1}$ in cy40NEW. A comparison of (11) against LES diagnosed entrainment rates is presented in Fig. 6 of de Rooy et al. (2013) and reveals a reasonably good correspondence, especially in comparison with estimates following a Kain Fritsch type of formulation (Kain and Fritsch, 1990) as shown in Fig. 5 of de Rooy et al. (2013). 
Herewith all entrainment rates in the dual mass flux scheme are defined.

### 2.2.2  The mass flux profile

The counterpart of entrainment is detrainment, $\delta$, describing outflow of updraft air into the environment, see Eq. (8). Together with entrainment, the detrainment determines the change of mass flux with height. The mass flux profile is important as it e.g. determines where the properties of the updraft are deposited in the environment. Besides, mass flux is used as input for the 
turbulence and cloud scheme (sections 2.3 and 2.4).

Equation (8) is not applied for the dry updraft where area fraction is assumed to be constant, so applying the vertical velocity Eq. (7) suffices to solve $\mathcal{M}$. Consequently, dry updraft mass flux simply varies with its updraft vertical velocity (like in N09).

For the moist updraft we use the commonly applied mass flux closure at cloud base (Grant, 2001) :

$$\mathcal{M}_{z_{lcl}} = c_b w_* \tag{12}$$

with where $\mathcal{M}_{z_{lcl}}$ is the mass flux at cloud base and $w_*$ is the usual convective velocity scaling derived from the surface buoyancy flux and using the cloud base as the boundary later depth (Grant, 2001). Further $c_b$ is a constant, set to $0.03$ in





cy40REF (according to Grant (2001)) and to 0.035 in cy40NEW (following Brown et al. (2002)). In the sub-cloud layer the moist updraft mass flux is imposed to increases linearly to the value at cloud base.

In the cloud layer, variations in the mass-flux profile from case to case and hour to hour, can be almost exclusively related to variations in the fractional detrainment as first pointed out by de Rooy and Siebesma (2008) (from hereon RS08). This is supported by numerous LES studies (e.g. Jonker et al. (2006); Derbyshire et al. (2011); Böing et al. (2012); de Rooy et al. (2013)). Apart from empirical evidence, the much larger variation in $\delta$ and its strong link to the mass flux is explained by theoretical considerations in de Rooy and Siebesma (2010). Variations in $\delta$ partly arise from variations in cloud layer depth.

This aspect is taken care of by evaluating and prescribing mass flux with a non-dimensionalised height, $\hat{z} = \frac{(z-z_{\mathrm{lcl}})}{h}$ and mass flux, $\hat{m} = \frac{\mathcal{M}_{\mathrm{u}}}{\mathcal{M}_{z_{\mathrm{lcl}}}}$. Where $h$ is the cloud layer depth, $z_t - z_{\mathrm{lcl}}$, as diagnosed by the moist updraft. Here $z_{\mathrm{t}}$ is the top of the cloud layer defined where $w_{\mathrm{u,moist}}$ becomes $0\,ms^{-1}$ and $z_{\mathrm{lcl}}$ corresponds to the cloud base height. Variations in the shape of the non-dimensionalised mass flux profile related to environmental conditions, like vertical stability and relative humidity, can be well described by a $\chi_{\mathrm{crit}}$ dependence (RS08).

$$\hat{m}_* = c_1 \langle \chi_{\mathrm{crit}} \rangle_* - c_2 \tag{13}$$

where $\hat{m}_*$ is the non-dimensionalised mass flux in the middle of the cloud layer (RS08) and $\chi_{\mathrm{crit}}$ is the fraction of environmental air necessary to make updraft air just neutrally buoyant (Kain and Fritsch, 1990). The symbol $\langle \rangle_*$ denotes the average from cloud base to the middle of the cloud layer. So $\langle \chi_{\mathrm{crit}} \rangle_*$ represents environmental conditions the updraft experiences along its rise up to the middle of the cloud layer. Note that apart from environmental conditions, also the buoyancy of the updraft itself

determines $\chi_{\mathrm{crit}}$ (RS08). As discussed in RS08, (13) describes a physically plausible relationship: "Large values of $\langle \chi_{crit} \rangle_*$ can be associated with large clouds (of large radii) with high updraft velocities that have large buoyancy excesses and/or clouds rising in a friendly, humid environment". For small $\langle \chi_{crit} \rangle_*$ values the opposite can be expected. As discussed in RS08, updraft excess in LES (depending on sampling method) and in the model parameterisation will differ. Therefore, $\chi_{\mathrm{crit}}$ values in LES and model will differ, and consequently will the optimal constants in (13). We apply $c_1 = 5.24$ (conform LES, RS08) and

$c_2 = 0.39$. In addition, we limit $\hat{m}_*$ between 0.05 (strongly decreasing mass flux) and 1 (no net decrease in mass flux). The upper boundary can be reached in stratocumulus layers where $\chi_{\mathrm{crit}}$ values can be high due to a high humidity environment.

With $\hat{m}_*$ known, and under the assumption that $\delta$ is constant with height (see e.g. RS08) and that the entrainment varies as $z^{-1}$, the mass flux profile can be determined (for details see RS08). The shape of the mass flux profile can vary from convex to concave up to the middle of the cloud layer, from there mass flux decreases linearly to 0 at cloud layer top. A strong

support for (13) can be found in Böing et al. (2012). Based on 90 LES runs covering a wide variety of relative humidity and stability of the environment, Böing et al. (2012) revealed a strong correlation of LES mass flux profiles with (13).





## 2.3 Turbulence scheme

In cycle 36 and older versions, HARMONIE-AROME made use of the CBR turbulence scheme (Cuxart et al. (2000), Seity et al. (2011)). As discussed by de Rooy (2014) and B17 some model deficiencies can be related to the CBR scheme, most notably lack of cloud top entrainment. Therefore, turbulence scheme HARATU (HArmonie with RAcmo TUrbulence) was implemented. HARATU is based on a scheme originally developed for regional climate model RACMO (van Meijgaard et al., 2012) and is described in detail in Lenderink and Holtslag (2004), from hereon LH04. In comparison with LH04 some modifications were implemented in HARMONIE-AROME (see B17), mainly to ameliorate wind speed forecasts during stormy conditions. With HARATU, HARMONIE-AROME substantially improved on several aspects, especially wind speed (B17, de Rooy et al. (2010), de Rooy et al. (2017)). On the other hand, together with updates of other parameterisations, HARATU contributed to the underestimation of low cloud cover and overestimation of cloud base height. Both output parameters are crucial for e.g. aviation purposes and eliminating these two specific shortcomings became top priority in the Hirlam consortium.

A full description of the turbulence scheme can be found in LH04 and B17 but for convenience we here introduce the components and parameters involved in the adjustments. In our turbulence scheme, the eddy diffusivity (see Eq. (4)) is formulated as $K = l\sqrt{TKE}$. The length-scale formulation in HARATU essentially consists of two length scales: one for (strongly) stable conditions $l_{\mathrm{s}}$, and one for weakly stable and unstable conditions, $l_{\mathrm{int}}$. The latter, so-called integral length scale provides a "quadratic profile" for unstable conditions in the convective boundary layer, and is also matched to surface similarity near neutral conditions. For more stable conditions the common formulation

$$l_{\mathrm{s}} = c_{\mathrm{m,h}} \frac{\sqrt{TKE}}{N} \tag{14}$$

is used, where $c_{\mathrm{m,h}}$ is a constant for momentum or heat, $TKE$ is the turbulent kinetic energy and $N$ is the Brunt Vaisala frequency.

To get the final length scale $l_{\mathrm{m,h}}$ for all stability regimes as applied in (4) we need to interpolate between the different length scales.The need for this arises because the different length scales do not match very well in the intermediate stability regimes; for example, the stable length scale approaches infinity for neutral stability. For this interpolation the following ad-hoc form is used:

$$\frac{1}{l_{\mathrm{m,h}}^{\mathrm{p}}} = \frac{1}{\{\sqrt{(l_{\mathrm{int}}^2 + l_{\mathrm{min}}^2)}\}^{\mathrm{p}}} + \frac{1}{l_{\mathrm{s}}^{\mathrm{p}}} \tag{15}$$

where $l_{\mathrm{min}}$ is a minimum length scale

$$\frac{1}{l_{\mathrm{min}}} = \frac{1}{l_{\infty}} + \frac{1}{0.5 c_{\mathrm{n}} \kappa z} \tag{16}$$

with $c_{\mathrm{n}}$ is a constant and $\kappa$ is the von Karman constant. Note that, close to the surface, the length scale is limited to half the neutral length scale, $c_{\mathrm{n}}\kappa z$. Equations (15) and (16) are needed to interpolate smoothly between the stable length scale and





the integral length scale near the surface, and to provide a limit length scale for the free troposphere. We note that the square root term in Eq. 15 is in practice similar to taking the maximum of $l_{\text{int}}$ and $l_{\text{min}}$, which is for instance needed to provide a background length scale for the free troposphere above the boundary layer where the integral length scale will be small or zero.

For most parameters in the length scale formulation there is some theory that provides a reasonable range of values (LH04), but $l_{\infty}$ is a tuning parameter and likewise the interpolation method is ad-hoc based. In LH04 an inverse lineair ($p = 1$) but also an inverse quadratic ($p = 2$) interpolation is discussed. In cy40REF an inverse lineair interpolation is used which suppresses mixing over a broad range of stability conditions. While the chosen form provides reasonably smooth transitions between the different stability regimes, results are sensitive to the interpolation and chosen constants, e.g. for $l_{\infty}$, and this will be investigated in section 3.2. Although the appropriate value for $l_{\infty}$ is uncertain, this parameter significantly influences the entrainment flux and hence the preservation of the inversion at the top of the boundary layer (section 3.4). The role of $l_{\text{min}}$ resembles that of the free tropospheric length scale mentioned by Bechtold et al. (2008) and Kőhler et al. (2011), who demonstrate the impact on inversion strength and consequently erosion of stratocumulus.

The last aspect of the turbulence scheme we discuss concerns the sub-cloud cloud interaction. The massflux contribution to the total vertical transport results in a stable stratification in the upper part of the sub-cloud layer. Consequently, mixing by the TKE scheme will be strongly diminished in this area. These feedbacks between the mass flux and the turbulence scheme generally lead to an unrealistically strong inversion at cloud base. In many mass flux schemes this runaway process is prevented by numerical diffusion which is dependent on the vertical resolution, and results of these schemes therefore tend to break down at very high resolution (Lenderink et al., 2004). For this reason an ad-hoc additional diffusion with constant $50 \cdot M_{\text{moist}}$ was added in cy40REF. In cy40NEW we replaced this term with a more physically based energy cascade term.

Let us briefly discuss the underlying ideas of the energy cascade term. Its formulation is inspired by the prognostic Eq. of the mass flux vertical velocity variance (de Roode et al. (2000) Eq. 2.12 for $w$):

$$\frac{\partial a_{\text{u}}(1 - a_{\text{u}})(w_{\text{u}} - w_{\text{env}})^2}{\partial t} = -2\mathcal{M}_{\text{u}}(w_{\text{u}} - w_{\text{env}})\frac{\partial \overline{w}}{\partial z} - \frac{\partial(1 - 2a_{\text{u}})\mathcal{M}_{\text{u}}(w_{\text{u}} - w_{\text{env}})^2}{\partial z}$$
$$- (\varepsilon + \delta)\mathcal{M}_{\text{u}}(w_{\text{u}} - w_{\text{env}})^2 + 2a_{\text{u}}(1 - a_{\text{u}})(w_{\text{u}} - w_{\text{env}})(S_{w_{\text{u}}} - S_{w_{\text{env}}}) \tag{17}$$

Here $S$ represents source terms and $w_{\text{env}}$ is the vertical velocity of the updraft environment. Since for convective clouds $\|w_{\text{env}}\| \ll w_{\text{u}}$, $w_{\text{env}}$ is, as usually, neglected. The LHS of 17 represents the change of the organised (or updraft) vertical kinetic energy. The third term on the RHS, representing the impact of lateral mixing, is always a negative or sink term and can be related to the energy cascade from organised to smaller-scale eddies. We apply this term as a source in the TKE budget equation. However, considering the increased complexity of having two updraft types and to prevent too high TKE values in the sub-cloud layer, we implemented the energy cascade term in an ad-hoc, simplified form:

$$W_{\text{casc}} = W_{\text{casc,dry}} + W_{\text{casc,moist}} = c\varepsilon_{\text{dry}}w_{\text{u,dry}}^2\mathcal{M}_{\text{dry}} + Fw_{\text{u,moist}}^2\mathcal{M}_{\text{moist}} \tag{18}$$

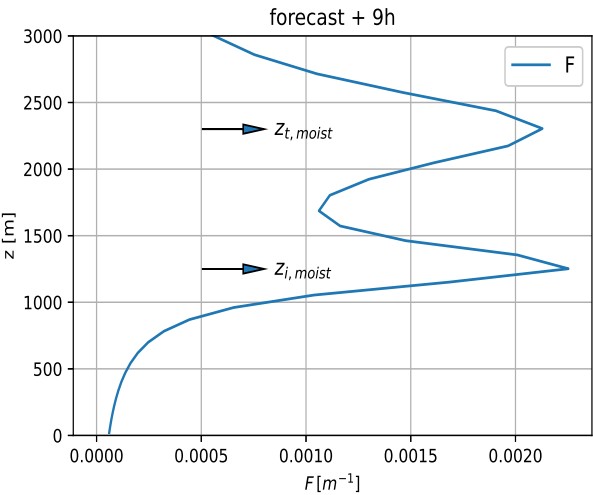

**Figure 3.** Profile of $F$ Eq. (19) at the 9th simulation hour of the ARM case (section 3.1) with cy40NEW.

with function F:

$$F = E_\mathrm{l}\left(\frac{z}{z_\mathrm{i}}\right)\left(\frac{1}{1+\left(\frac{z_\mathrm{i}-z}{Z_\mathrm{wl}}\right)^2}\right) + E_\mathrm{t}\left(\frac{1}{1+\left(\frac{z_\mathrm{top}-z}{Z_\mathrm{wt}}\right)^2}\right) \tag{19}$$

here $c = 0.5$, $Z_\mathrm{wl} = 200m$, $Z_\mathrm{wt} = 400m$. Further, $E_\mathrm{l} = 0.002m^{-1}$ is a typical $\varepsilon$ value near cloud base (consistent with Eqs. (10) and (11)) and $E_\mathrm{t} = 0.002m^{-1}$ corresponds to a similar peak at the level of neutral buoyancy but this time associated with detrainment in the upper part of the cloud layer. Fig. 3 shows a typical profile of (19). By ignoring the detrainment term in the

dry updraft contribution (Eq. (18)) and applying function $F$ (Eq. (19)) for the moist updraft, too large TKE values in the lower part of the boundary layer are prevented whereas the contribution to TKE near cloud base and in the upper part of the cloud layer is supported.

Next to the usual dissipation, transport, buoyancy and shear term, $W_\mathrm{casc}$ is added as a source term in the $TKE$ budget equation. LES results in section 3.1.1 substantiate the need for the energy cascade term and demonstrate the improved

turbulent transport in cy40NEW due to the inclusion of the energy cascade term.

## 2.4 Statistical cloud scheme

Accurate predictions of clouds, liquid water and ice are important because they have a large impact on radiation and therewith on several components of the model. This applies in particular to low, boundary layer clouds such as stratocumulus and cumulus. In HARMONIE-AROME high (ice) clouds are parameterised separately in a relative humidity scheme (B17) and are





outside the scope of this paper. The here presented derivations, ideas and modifications concerning parameterisation of low clouds in HARMONIE-AROME are valuable for statistical cloud schemes in general.

The concept of parameterising clouds with a statistical cloud scheme was already pioneered by Sommeria and Deardorff (1977) and Mellor (1977) and makes use of the fact that cloud cover and liquid water content can be easily derived once sub-grid variability of moisture and temperature are known. This concept has been further developed by Bougeault (1981) by
assuming specific analytical forms of the joint probability functions (PDF's) of total water specific humidity $q_t$ and liquid water potential temperature $\theta_l$, which are the relevant thermodynamic moist conserved variables. From several successive papers (Bechtold et al. (1995), Cuijpers and Bechtold (1995), Bechtold and Siebesma (1998)) it became clear that it is sufficient to have reliable estimates of only the grid box variances of $q_t$ and $\theta_l$ without making explicit assumptions on the shape of the underlying PDF.

In statistical cloud schemes relevant information on $q_t$ and $\theta_l$ is captured in one variable called $s$, distance to the saturation curve, $s \equiv q_t - q_s$ with $q_s$ being the saturation specific humidity. If we non-dimensionalise $s$ by its standard deviation $\sigma_s$, $t \equiv s/\sigma_s$, and presume a Gaussian PDF for $t$, the cloud fraction and liquid or ice water content can be written as a function depending only on the mean value of $t$:

$$\overline{t} = (\overline{q}_t - \overline{q}_s)/\sigma_s \tag{20}$$

Because $\overline{q}_t - \overline{q}_s$ is readily available in a model, the cloud parameterisation problem is simply reduced to estimating $\sigma_s$.

The base of statistical cloud schemes is an expression of variance in $s$ in terms of variances and covariance of $q_t$ and $\theta_l$. Although the exact notation might be different, this expression should be the same for all schemes because the derivation is based on fundamental thermodynamics. Nevertheless, erroneous solutions can be found in literature as well as in cy40REF. Therefore, we provide a step by step derivation of the variance in $s$ in appendix A1, which finally results in the following
expression:

$$\sigma_s^2 = \overline{s'^2} = \alpha^2 \overline{q_t'^2} - 2\alpha^2 \beta \overline{q_t'\theta_l'} + \alpha^2 \beta^2 \overline{\theta_l'^2} \tag{21}$$

with

$$\alpha = \frac{1}{1 + \frac{L}{c_p} q_{sl,T}}, \quad \beta = \pi q_{sl,T} \tag{22}$$

$$\overline{q}_{sl,T} = \frac{\partial q_s(\overline{T}_l)}{\partial T} \tag{23}$$

using the definition of the liquid water temperature:

$$T_l \equiv T - \frac{L}{c_p} q_l, \tag{24}$$


and where $L$ is the latent heat of vaporization and $c_{\mathrm{p}}$ the heat capacity of dry air at constant pressure, and $\pi$ is the Exner function defined as $\pi = (\frac{p}{p_0})^{\frac{R_{\mathrm{d}}}{c_{\mathrm{P}}}} = \frac{T}{\theta}$ in which $R_{\mathrm{d}}$ is the gas constant of dry air and $p_0$ a reference surface pressure.

In literature, several approaches exist to estimate $\sigma_{\mathrm{s}}$ (e.g. Golaz et al. (2002), Bechtold et al. (1995)). Here we provide a full description of our estimate in which we include the contribution to the variance by turbulence and convection as well as an additional term to cover other sources of variance.

If we neglect advection, precipitation and radiation terms, the budget equations for (co)variances are (see e.g. Stull (1988)):

$$
\quad \frac{\partial \overline{a'b'}}{\partial t} = -\frac{\partial \overline{w'a'b'}}{\partial z} - [\overline{w'a'}\frac{\partial \overline{b}}{\partial z} + \overline{w'b'}\frac{\partial \overline{a}}{\partial z}] - \epsilon_{\mathrm{ab}} \tag{25}
$$

where $a,b \in \{\theta_{\mathrm{l}}, q_{\mathrm{t}}\}$. The first term on the RHS of (25) is the transport term, the second and third term represent the impact of the turbulent fluxes and the last term covers dissipation. According to Bechtold et al. (1992) the transport term can be neglected during conditions with substantial cloud cover. The dissipation term, $\epsilon_{\mathrm{ab}}$ is modelled by a Newtonian relaxation back to isotropy:

$$
\quad \epsilon_{ab} = \epsilon_{\mathrm{ab,turb}} + \epsilon_{\mathrm{ab,conv}} = c_{\mathrm{ab}}(\frac{\overline{a'b'}^{\mathrm{turb}}}{\tau_{\mathrm{turb}}}) + c_{\mathrm{ab}}(\frac{\overline{a'b'}^{\mathrm{conv}}}{\tau_{\mathrm{conv}}}) \tag{26}
$$

where $c_{\mathrm{ab}}$ is a constant and $\tau$ is a time scale for dissipation of turbulence ($turb$) or convection ($conv$). It is not clear if $c_{\mathrm{ab}}$ should be different for turbulence and convection. Moreover, a large variation in its value can be found in literature (see e.g. Bechtold et al. (1992), Redelsperger and Sommeria (1981)). For turbulence $\tau$ can be approximated by

$$
\tau_{\mathrm{turb}} = \frac{l_\epsilon}{\sqrt{TKE}} \tag{27}
$$

where $l_\epsilon = l_{\mathrm{m}}c_0^2$ is the dissipation length scale with $c_0 = 3.75$ (see LH04, and consistent with the turbulence scheme). In cy40REF however, $l_\epsilon = l_{\mathrm{m}}$ (discussed in section A2). The time scale for convection can be related to the cloud depth divided by a typical cumulus updraft velocity (Lenderink and Siebesma, 2000). However, for simplicity we adopt the approach of Soares et al. (2004) taking $\tau_{\mathrm{conv}} = 600s$.

Similar to dissipation, the turbulent fluxes in (25) consist of diffusive transport covered by the turbulence scheme

$$
\quad \overline{w'a'} = -K\frac{\partial \overline{a}}{\partial z} = -l_{\mathrm{m,h}}\sqrt{TKE}\frac{\partial \overline{a}}{\partial z} \tag{28}
$$

where all stability factors are included in length scale $l_{\mathrm{m,h}}$ (LH04), and convective transport by the mass flux scheme:

$$
\overline{w'a'} = \mathcal{M}_{\mathrm{u}}(a_u - \overline{a}) \tag{29}
$$

As mentioned above, we neglect the transport term in (25) and assume a steady state, i.e. the LHS of (25) is 0. This means that production and dissipation of (co)variances are in balance. Note that the steady state assumption is, at least for convection,





debatable because the timescale for dissipation of convection is an order of magnitude larger than the typical time step of our model. On the other hand, cloud fractions for shallow, unresolved convection are usually small. Because we consider contributions of both turbulence and convection to the variance, we assume a balance between production and dissipation for both processes separately. Substituting (26), (28), (29), $\tau_{\text{turb}}$ and $\tau_{\text{conv}}$ in (25), including the assumptions mentioned above, leads to the following expressions:

$$\overline{a'b'}^{\text{turb}} = 2\frac{l_{\text{m,h}}l_{\varepsilon}}{c_{\text{ab}}}\frac{\partial \overline{a}}{\partial z}\frac{\partial \overline{b}}{\partial z} \tag{30}$$

$$\overline{a'b'}^{\text{conv}} = \frac{-\tau_{\text{conv}}}{c_{\text{ab}}}\left(\mathcal{M}_{\text{u}}(a_{\text{u}}-\overline{a})\frac{\partial \overline{b}}{\partial z} + \mathcal{M}_{\text{u}}(b_{\text{u}}-\overline{b})\frac{\partial \overline{a}}{\partial z}\right) \tag{31}$$

So for example total variance in $\theta_{\text{l}}$ due to turbulence and convection reads:

$$\overline{\theta_l'^2} = 2\frac{l_{\text{m,h}}l_{\epsilon}}{c_{\text{ab}}}(\frac{\partial \overline{\theta_l}}{\partial z})^2 - \frac{2\tau_{\text{conv}}}{c_{\text{ab}}}(\mathcal{M}_{\text{u}}(\theta_{\text{l,up}}-\overline{\theta_l})\frac{\partial \overline{\theta_l}}{\partial z}) \tag{32}$$

Note that both turbulence and convection have a positive contribution to variance.

In the absence of convection and no noticeable amount of turbulent activity, variance will still be non-zero. In nature other sources of variance exist like surface heterogeneity, horizontal large-scale advection, meso-scale circulations and gravity waves. Instead of imposing a minimum value to variance to cover these sources, we apply an extra variance term with the characteristics of a relative humidity scheme. This additional term was already introduced in de Rooy et al. (2010) demonstrating its beneficial impact, and included in the HARMONIE-AROME reference code since cycle 36. Here a more elaborated description of the additional variance term is given.

Let us assume a statistical cloud scheme with a uniform distribution of a fixed width $2\Delta$. Tompkins (2005) shows that such a statistical cloud scheme can be considered as a RH scheme with:

$$\Delta = (1 - RH_{\text{crit}})q_{\text{s}} \tag{33}$$

with $RH_{\text{crit}}$ representing the relative humidity where cloud fraction starts to be non-zero. The corresponding cloud fraction reads:

$$a_{\text{c}} = 1 - \sqrt{\frac{1-RH}{1-RH_{\text{crit}}}} \tag{34}$$

The variance of such a uniform distribution is:

$$\sigma_{q_{\text{t}}}^2 = \frac{1}{3}\Delta^2 \tag{35}$$

Tompkins (2005) and Quaas (2012)) demonstrated that a RH-scheme as well a statistical cloud scheme with a fixed width distribution could be written purely in terms of specific humidity fluctuations, i.e. Eq. (21) reduces to:

$$\sigma_{\text{s}}^2 = \overline{s'^2} = \alpha^2\overline{q_{\text{t}}'^2} = \alpha^2\sigma_{\text{qt}}^2 \tag{36}$$





Combination of (33), (35) and (36) leads to the following expression for $RH_{\mathrm{crit}}$:

$$RH_{\mathrm{crit}} = 1 - \frac{\sqrt{3}}{\alpha}\left(\frac{\sigma_{\mathrm{s}}}{q_{\mathrm{s}}}\right) \tag{37}$$

In HARMONIE-AROME we introduced the additional standard deviation term

$$\sigma_{\mathrm{s,extra}} = c\alpha q_{\mathrm{s}} \tag{38}$$

With $c = 0.02$ this leads to a constant $RH_{\mathrm{crit}} = 96\%$ (37). Note that due to pre-factor $\alpha$ in (38), $RH_{\mathrm{crit}}$ becomes independent of $\alpha$. For typical atmospheric conditions $\alpha \simeq 0.4$ in the boundary layer, while higher up in the atmosphere $\alpha$ will asymptote towards unity. Therefore, without pre-factor $\alpha$ in (38), $RH_{\mathrm{crit}}$ would vary from $\simeq 91\%$ in the boundary layer to $\simeq 96\%$ in the
upper atmosphere. However, sources of variance, not related to turbulence or convection, are particularly found higher up in the atmosphere (see e.g. Quaas (2012)) and are e.g. related to advection of long lived cirrus clouds into the model grid box. Therefore, $RH_{\mathrm{crit}}$ should at least not increase with height. More investigation is needed to optimise the (height dependent) formulation of the additional variance term. The total variance in $s$ is the sum of the contributions from turbulence, convection and (38).

From the description above and Appendix A, it becomes clear that a statistical cloud scheme contains many uncertain terms and constants. We do not claim that our choices are all optimal. However, in comparison with the original scheme, the new set-up is at least build upon a correct derivation of the thermodynamical framework. This is e.g. important for the formulation of thermodynamic coefficients (22). Therefore, we believe the new set-up is more suitable as a starting point for further improvements.

## 3   Argumentation and evaluation of model updates

This paper describes a large variety of modifications to the current reference cloud, turbulence and convection parameterisations. Argumentation of these adjustments is diverse. For example, part of the changes to the cloud and turbulence scheme have a theoretical basis, namely thermodynamics and surface layer similarity, respectively. Other modifications are substantiated by an in-depth comparison of 1D model results with LES for several idealised inter-comparison cases. Lastly, optimisation of
some more uncertain model parameters is based upon evaluation of full 3D model runs. Considering the large number of modifications and mutual influences, it is impossible to discuss the separate and incremental impact of them all. Instead we focus on the performance of two HARMONIE-AROME configurations: firstly, the reference HARMONIE-AROME set-up as described in B17, cy40REF, and, secondly, the new configuration, cy40NEW, as proposed in this paper. Nevertheless, all adjustments are substantiated and the isolated impact of several of them is demonstrated. An overview of all modifications is presented in Table
D1 in Appendix D.





Many of the proposed adaptations are the result of a comparison of 1D model with LES results as obtained with the DALES model (Heus et al., 2010). For an accurate comparison between LES and HARMONIE-AROME at the current model resolution, LES results are diagnosed as the mean over Harmonie-sized sub-domains. In the ARM shallow cumulus case for example, the turbulent transport in LES is the mean turbulent transport diagnosed in 100 sub-domains of $2.5 \times 2.5$ km$^2$, the

current operational resolution of HARMONIE-AROME. However, differences between the mean over Harmonie-sized sub-domains and the mean across the full LES domain are generally small. We start in section 3.1 with an elaborated comparison of 1D model with LES results for the ARM case. This investigation involves many components of the parameterisations and several modifications are based on the ARM case. By making use of Monin-Obukhov theory (following Baas et al. (2017)), important changes to the turbulence scheme are substantiated in section 3.2. This section also shows the performance under

moderately stable conditions in the GABLS1 case (Beare and M.K. Macvean, 2006). Section 3.3 mainly demonstrates the impact of the modifications on three stratocumulus cases. Finally, long term and case based verification with the 3D model is presented in section 3.4. This section demonstrates the large improvement with the updates in cy40NEW on low clouds but also elucidates the beneficial impact on precipitation.

### 3.1 ARM case

The ARM case (Brown et al., 2002), based on observations, describes a diurnal cycle of shallow convection above land: initiation of moist convection, gradual deepening of the cloudy layer and finally collapse of the cumulus cloud layer. Such a dynamical case poses higher demands to convection parameterisation than e.g. the steady-state BOMEX case over sea (Holland and Rasmusson, 1973) and is therefore more suitable for optimisation purposes. To make optimal use of the dynamical character of the ARM case and to avoid cherry picking, we present results of all hours during the moist convective period (simulations

from $+4$ to $+12$ h).

#### 3.1.1 ARM: Mass flux and total turbulent transport

With the current operational resolution of HARMONIE-AROME, turbulent transport in the ARM case is fully unresolved and is presented as the sum of parameterised convective and diffusive turbulent transport. In LES however, shallow convection and the bulk part of the diffusive transport is resolved. By sampling LES data in the cloud layer we can estimate that part of the total turbulent transport that should be described by a convection scheme. Although the convective transport by LES

should be interpreted as a rather crude estimate, it is also the best available way to study the performance of our mass flux convection scheme in the cloud layer. A detailed description of such an evaluation is provided in Appendix B and indicates that the convective transport in HARMONIE-AROME is underestimated in the first half of the convective period in the ARM case but modifications to the convection scheme in cy40NEW result in a clear reduction of this underestimation (Appendix B).







**Figure 4.** Total turbulent transport $\left[\frac{m}{s}\right]$ of the mixing ratio total humidity $r_\mathrm{t}$ during all convective hours of the ARM case, corresponding to simulation hours +4 to +12 hours. Plotted is total turbulent transport of cy40REF (orange), cy40NEW (green) and the total turbulent transport by the LES (blue). Note that the x-axis scale is not constant.

However, the ultimate goal of a convection and turbulence scheme is to provide an accurate estimate of the total turbulent transport. After all, the vertical divergence of the total turbulent transport determines the tendencies of the prognostic model variables. Whereas LES convective transport should be interpreted as an estimate, depending on the sampling method, LES total turbulent transport during the ARM case will be close to observed values. Besides, in contrast to convective transport, LES provides the total turbulent transport for the complete atmosphere, including the sub-cloud layer. Fig. 4 shows the total

turbulent transport of humidity by the model versions and LES, including the LES subgrid-scale, parameterised contribution. Plots of heat transport provide a similar behaviour (not presented). In general, both model versions underestimate total turbulent transport but the new configuration results in a considerable improvement. Drying of the sub-cloud layer, i.e. increasing total turbulent transport with height, in the second half of the convective period is almost absent in the original configuration and better captured with cy40NEW. This improvement is mainly related to inclusion of the energy cascade (19) as demonstrated



**Figure 5.** The kinematic total turbulent transport $\left[\frac{m}{s}\right]$ during the last 4h of the ARM convective period. Plotted is the transport according to LES (blue), cy40NEW (green) and cy40NEW but without energy cascade (green dashed). Note that the x-axis scale is not constant.

in Fig. 5. Fig. 5 further reveals that the energy cascade smoothens wiggles in turbulent transport around the inversion at cloud base. Fig. 6 shows the humidity profiles at the end of the convective period, therewith reflecting the accumulated impact of turbulent transport during the ARM case. There is a close agreement between the humidity profiles of Cy40NEW and LES whereas the cy40REF run clearly leads to a too moist sub-cloud and too dry cloud layer. As discussed before, especially the more efficient sub-cloud to cloud transport in cy40NEW is responsible for the large improvement in the humidity profile.

A closer examination of Figs. B1 and 4 reveals something remarkable: If we compare LES organised cloudy updraft transport (Fig. B1) with LES total turbulent transport (Fig. 4) in the upper part of the cloud layer, it becomes clear that organised transport alone would overestimate total transport in this region. If we look e.g. at the $+10h$ forecast, LES show almost no total turbulent transport above $2500m$ despite considerable convective transport. To investigate this we decompose the total turbulent transport in LES. Following Siebesma and Cuijpers (1995), total turbulent transport can be written as a sum of



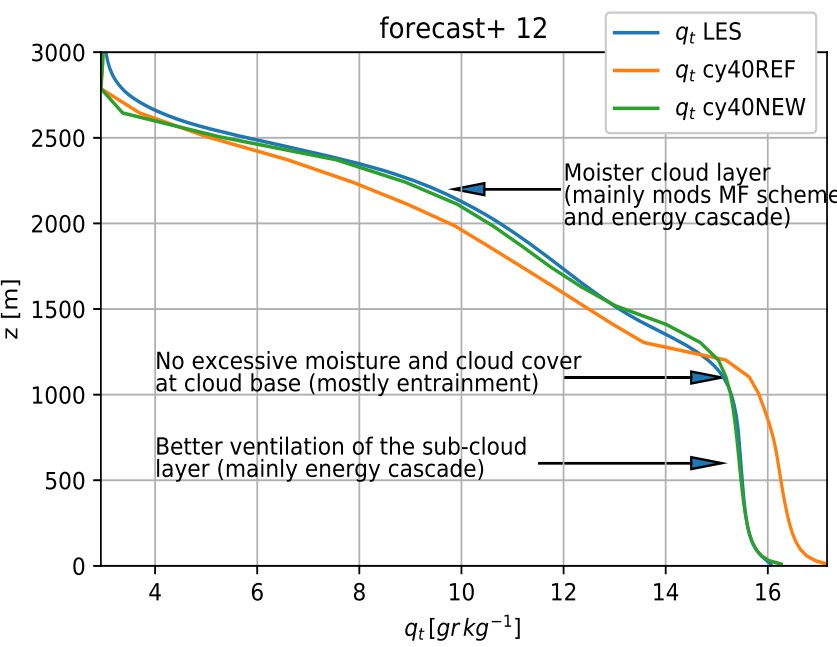

**Figure 6.** ARM case Specific humidity profile after 12 hours of simulation. These profiles can be seen as the accumulated impact of the total turbulent humidity transport during the ARM case.

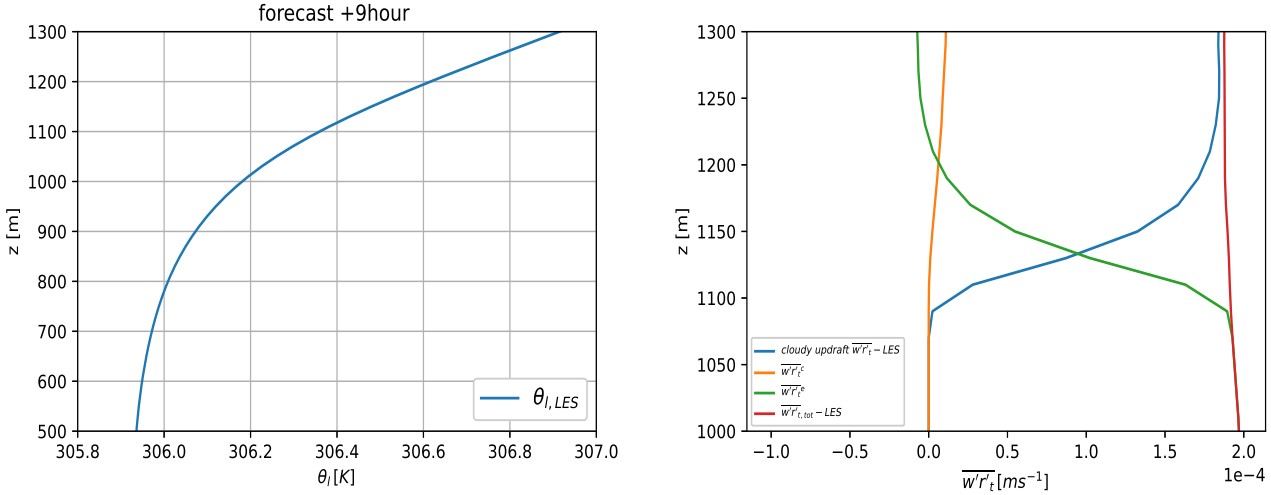

**Figure 7.** LES results around the cloud base inversion height for the ARM case at the 9th simulation hour. The left panel shows the $\theta_l$ profile whereas the right panel presents a decomposition of the kinematic turbulent moisture fluxes $\left[\frac{m}{s}\right]$. Plotted are LES cloudy updraft flux (blue), small-scale sub-plume transport (orange), small-scale environmental transport (green), and total transport (red). Note the different y-axis scale.





large-scale organised and small-scale sub-plume and environmental transport. In Appendix C we elaborate on the nature of the turbulent transport in the upper part of the cloud layer by examining decomposed terms of the turbulent transport. This examination reveals that the rather good approximation of the total turbulent transport in the upper part of the cloud layer by the parameterisation seems to be the result of a compensation error in the ARM case; too shallow mass flux transport is balanced by neglecting downward environmental turbulence (see Appendix C).

Additionally, the decomposition is used to look specifically into the turbulent transport around the cloud base inversion height in relation to the energy cascade term (18), see Fig. 7. As shown in the left panel of Fig. 7, the LES $\theta_l$ profile around $1000m$ height and after 9 simulation hours is roughly the stable lapse rate (without phase changes) of the cloud layer. Considering this atmospheric stability, a standard turbulence scheme would provide little mixing at this, and higher, levels. However, the right panel of Fig. 7 reveals that the total turbulent transport is actually dominated by (small-scale) diffusive

environmental turbulence up to considerably above the inversion height (in agreement with Figs. 7a and b in Siebesma and Cuijpers (1995) for the BOMEX shallow convection case). A plausible explanation for the presence of diffusive transport despite the stable conditions are (dry) updrafts terminating around the inversion height, in this way feeding the energy cascade from larger to smaller scales. In addition, organised entrainment at cloud base height (de Rooy and Siebesma, 2010) induced by acceleration of the moist updraft might further enhance small-scale environmental turbulence in this area. To describe this

important contribution to the transport from sub-cloud to cloud layer, the energy cascade term (18) is added (section 2.3).

     Based on this shallow cumulus case it is evident that the physical basis of our parameterisation is a strong simplification of reality. Moreover, the rather good approximation of the total turbulent transport during the ARM case is partly caused by a compensating error (Appendix C). However, a realistic representation would require a substantial increase in complexity, introducing new uncertain, tune-able parameters. Moreover, the current set of parameterisations performs well on a wide

variety of cases.

### 3.1.2   Cloud cover

A contour plot of cloud fraction during the ARM case (Fig. 8) reveals that cy40 NEW results in lower maximum cloud fraction (near cloud base) in better correspondence with LES. This is also reflected in reduced total cloud cover (Fig. 9). Figure 9 further reveals that observed maximum total cloud cover is higher than in LES and peaks earlier. Brown et al. (2002) argues that the

difference in timing between model results and observations is caused by differences between the initial profiles as prescribed in the case set-up and the observations.

     Observed differences in cloud fraction and cover between cy40REF and cy40NEW (resp. Figs. 8 and 9) are the accumulated result of several modifications:



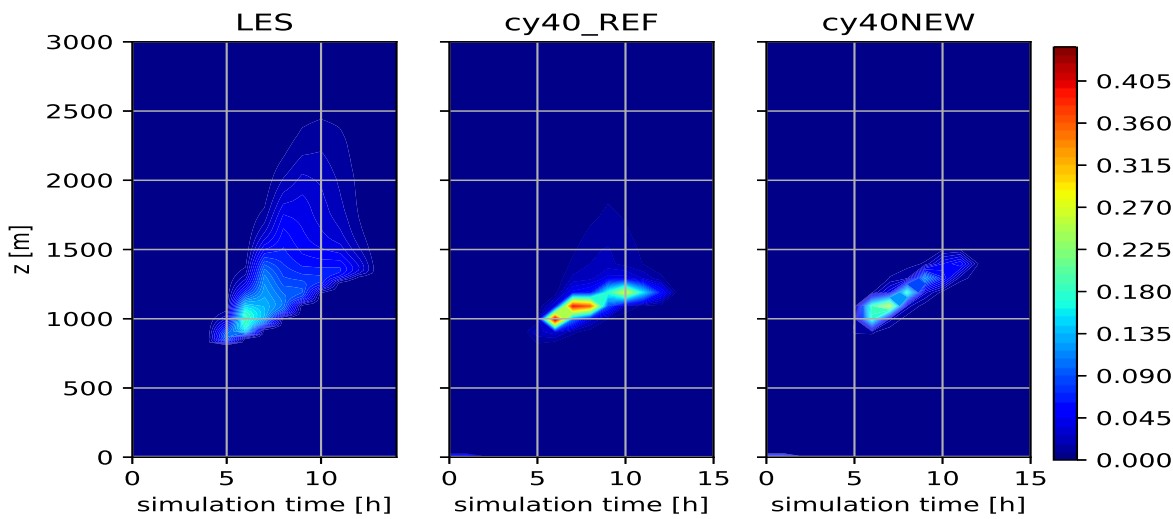

**Figure 8.** Contourplot of cloud fraction for the ARM case

- As illustrated in Figs. 5 and 6, the reference configuration underestimates ventilation of the boundary layer leading to
too high humidity values near cloud base and therefore too high maximum cloud fraction values. Especially the energy cascade (18) is responsible for the enhanced ventilation (section 3.1.1).

- Humidity near cloud base is also influenced by the dry updraft. In the reference formulation, Eq. (9) with $a_2 = 40$, entrainment, and therewith dilution of the updraft, remains rather small approaching the inversion. When this dry updraft finally terminates, relatively high amounts of moisture are detrained in the environment in cy40REF. With $a_2 = 1m$, as
in cy40NEW, this effect is mitigated.

- Another contribution to the different results stems from the removal of bugs in the reference cloud scheme. Most notably are erroneous thermodynamic coefficient $\beta$ (in Tudor and Mallardel (2004)) and double application of factor 2 on the contribution to the variance by convection (appendix A2). Especially the latter bug in cy40REF leads to a substantial increase in variance and accordingly to higher cloud fraction at cloud base.

- The largest impact is related to the choice of parameter $c_{ab}$ (section 2.4 Eq. (26) and appendix A2). If $c_{ab} = 1$ from cy40REF would be applied in the new configuration, the variance, and with it the cloud cover, would be substantially overestimated as demonstrated in Fig. 9. Only in cy40NEW $c_{ab}$ is in line with literature (Redelsperger and Sommeria, 1981), i.e. 0.139.

Apart from the (too) high cloud fractions at cloud base, also the underestimation of low values of cloud fraction in
the upper part of the cloud layer by both model versions stands out in Fig. 8. Because the humidity (see Fig. 6) and temperature (not shown) profiles of Cy40NEW closely match LES, the underestimation of cloud fraction in the upper part of the cloud layer





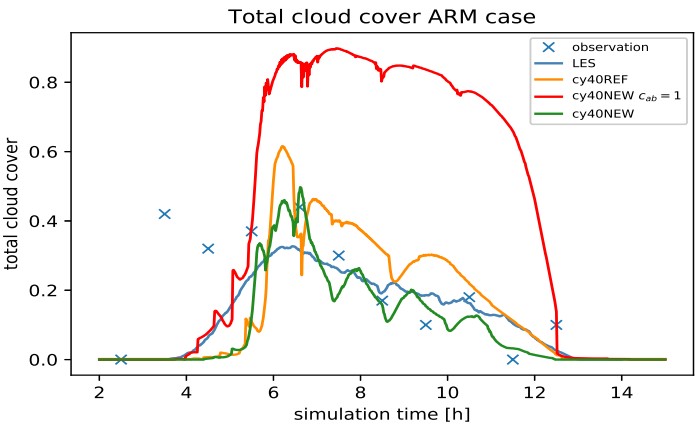

**Figure 9.** Total cloud cover ARM case. Plotted are observations (blue crosses), LES (blue), cy40REF(orange), cy40NEW with $c_{ab} = 1$ (red, see eqs. (26), (30), (31)) and cy40NEW (green)

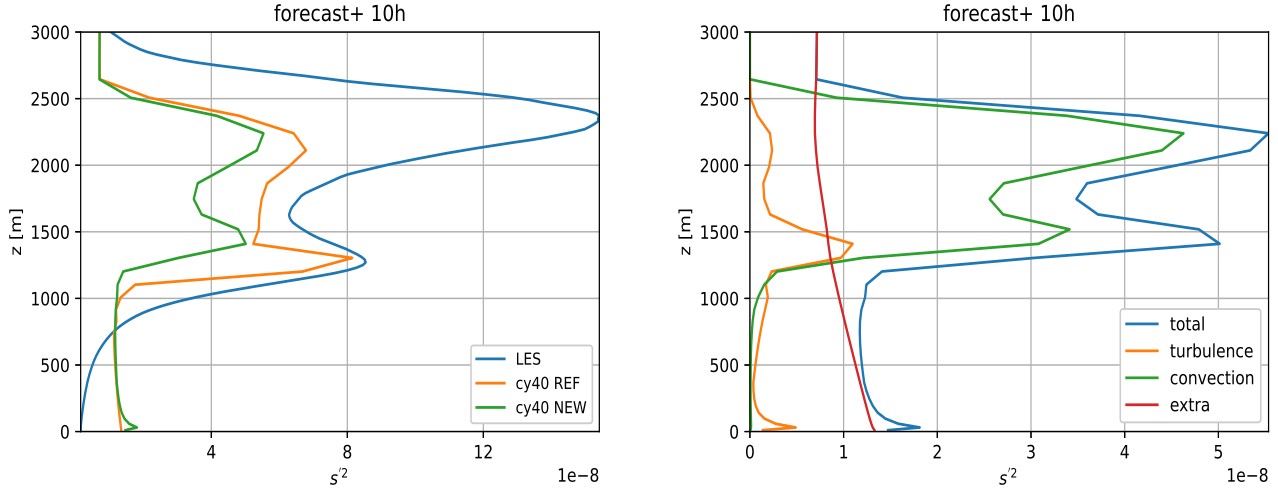

**Figure 10.** ARMcase, 10th simulation hour. Panel a) shows the profile of the variance in $s$ in LES (blue), cy40REF (orange) and cy40NEW (green). Panel b) shows the contribution of the turbulence (orange), convection (green) and the extra term (red) Eq. (38), to the variance in $s$ for cy40NEW

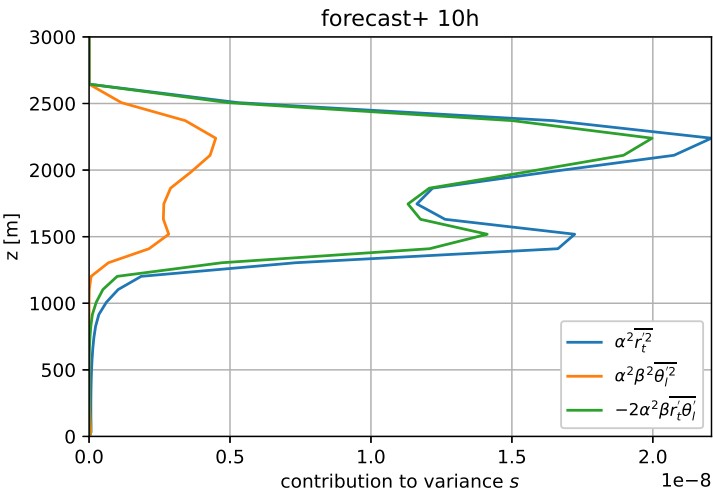

**Figure 11.** ARMcase, 10th simulation hour. The convective contribution to the variance in $s$ of cy40NEW from the variance in $\theta_l$ (orange), total mixing ratio $r_t$ (blue) and the co-variance (green), see Eq.(21).

must be related to an underestimation of variance in $s$. Figure 10a (for a typical hour) indeed reveals that both model versions underestimate the variance in $s$ in the cloud layer, although cy40REF values are closer to LES. While the new configuration generally improves the shape of the variance profile, the local maximum near cloud-top should be more pronounced. Note that

inclusion of the convective co-variance term, $\overline{r_t'\theta_l'}$ helps to increase the local maximum near cloud top (Fig. (11)). Figure 10b clearly demonstrates that the contribution of convection to the variance in $s$ is essential to adequately describe the shape of the variance profile in the cloud layer, especially the maximum near cloud top. Furthermore, it was decided not to include the contribution of the dry updraft to variance. First of all, together with the extra variance term (section 2.4, Eq. (38), Fig. 10b), variance in the lower half of the sub-cloud layer would be too high. Moreover, with fluctuations in the termination level of the

dry updraft, cloud cover near cloud base height changes, which can lead to noisy cloud cover patterns (not shown).

     Although the cloud scheme of cy40NEW already performs satisfactorily for a wide variety of weather conditions, there are clearly several options for further optimisation. Examples of possible improvement are: Introduction of a height dependence of the extra variance term, partial replacement of the extra variance term by a dry updraft contribution in the sub-cloud layer, increasing $\tau_{\mathrm{conv}}$ Eq. (31) because the current value (Soares et al., 2004) seems to be on the low side (compare to

e.g. Siebesma et al. (2003)), or modifying the energy cascade function (19) to increase the local maximum around cloud top. Nevertheless, with a more sound physical bases and the removal of bugs, the new cloud scheme set-up is already better suited as a base for such new developments.





## 3.2 Optimizing the turbulence scheme

Two important modifications in the turbulence scheme are based on an evaluation procedure as described by Baas et al. (2008)
and Baas et al. (2017). They demonstrated that a comparison of the dimensionless gradients of heat, $\phi_h$, and momentum,
$\phi_m$, versus the stability parameter, $\frac{z}{\Lambda}$ (39), enables a more physically-based choice of turbulence parameter settings for stable
conditions.

$$\frac{z}{\Lambda} = \frac{-\kappa z \frac{g}{\theta_v} \overline{w'\theta'_v}}{u_*^3} \tag{39}$$

where $\Lambda$ is the local Obukhov length and $u_*$ is the friction velocity. According to similarity theory there should be a universal
relation between the dimensionless gradients and the stability parameter, although the uncertainty in these relations increases
for stronger stratification, i.e. larger $\frac{z}{\Lambda}$ values.

To investigate the mixing characteristics of our turbulence scheme in terms of the similarity relations, a SCM of
HARMONIE-AROME is run for 1 year at the location of super observation site Cabauw (Bosveld et al., 2020). The SCM is
forced by output from daily three-dimensional forecasts of RACMO (van Meijgaard et al., 2008). Figure 12 shows the 1 year
SCM output diagnosed in terms of flux-gradient relations for momentum and heat. We present results with default cy40REF
settings, i.e. $p = 1$ (15) and $c_h = 0.15$ (14) next to $p = 2$ and $c_h = 0.11$ conform cy40NEW (see section 2.3). Evaluation
is restricted to stable boundary-layer regimes, i.e. positive values of $\frac{z}{\Lambda}$. Apart from model results also theoretical relations
according to Dyer (1974) in blue and Beljaars and Holtslag (1991) (green) and Duynkerke (1991) (yellow) are plotted. Many
observational studies on flux-gradient relations report that for increasing stability the exchange of momentum is far more
efficient than the exchange of heat, i.e. $\phi_h > \phi_m$ (see e.g. Beljaars and Holtslag (1991)). The relationship of Dyer (1974) does
not reflect this and we focus on the relations of Beljaars and Holtslag (1991) and Duynkerke (1991) that were both derived
from Cabauw observations. The divergence between the latter two flux-gradient relations for increasing stability illustrates
the uncertainty under very stable conditions (Baas et al., 2008). Therefore, most attention is paid to neutral to moderately
stable regimes, roughly corresponding with $0 < \frac{z}{\Lambda} < 1$. Figure (12) shows that in this stability range, the reference set-up
underestimates mixing (overestimates the gradient) which can be related to linear interpolation between the length scales,
i.e. $p = 1$. However, only changing interpolation to quadratic would lead to excessive mixing and unrealistic flux-gradient
relations (not shown). This can be compensated by reducing the proportionality factor of the stable length scale, $c_h$ to 0.11.
The combined result of these changes is shown in Fig. 12, where the lower panels reveal a better correspondence with the
flux-gradient relations in near neutral to moderately stable conditions. For more stable conditions agreement with theoretical
relations seems to deteriorate with the new set-up. However, as explained above the flux-gradient relations become highly
uncertain under these strongly stratified conditions. To explore the performance of the turbulence scheme in moderately stable
conditions, cy40REF and cy40NEW are compared to LES for the GABLS1 case (Beare and M.K. Macvean, 2006), based on
arctic observations. Although the change from $p = 1$ to $p = 2$ in the turbulence scheme (section 2.3 Eq. (15)) leads to increased



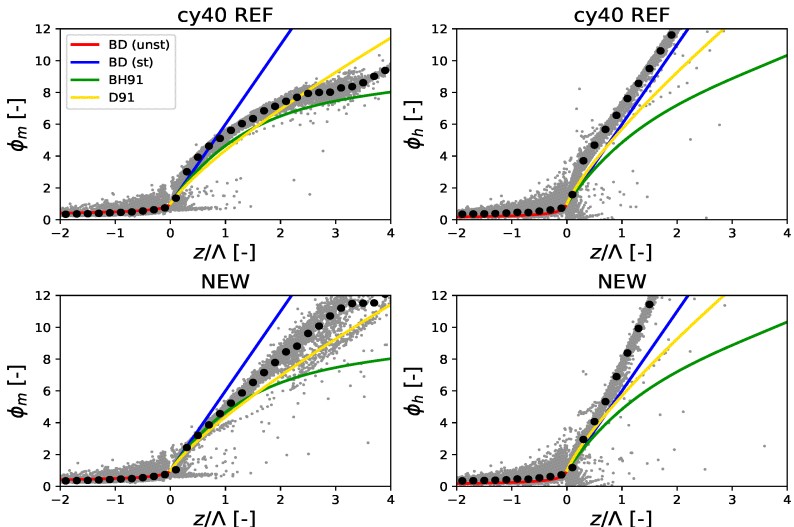

**Figure 12.** Dimensionless gradients of wind (left panels), and temperature (right panels), as a function of the local stability parameter $\frac{z}{\Lambda}$ as diagnosed from 1 year of SCM output (grey dots). The upper and lower panels show the results for cy40REF and cy40NEW respectively. Blue lines indicate $1 + 5\frac{z}{\Lambda}$ (Dyer, 1974), green lines (Beljaars and Holtslag, 1991) and yellow lines the relations proposed by Duynkerke (1991). Explanation of the different formulations can be found in the text. For completeness, Dyer (1974) formulations for unstable conditions are plotted (red line).

mixing in near neutral to weakly stable conditions, most other modifications, that reduce mixing (see section 2.3), dominate for
more stable conditions (see also Fig. 12). Results for GABLS1 (Fig. 13), showing the wind speed profile after $9h$ of simulation, indeed reveal more stable profiles and lower boundary layer heights with cy40NEW, in better correspondence with LES.

Due to increased mixing in near neutral conditions with $p = 2$, the updates in HARATU to increase momentum mixing in strong wind conditions (see B17), are removed. Removing these updates together with the reduced $c_{\mathrm{h}}$ coefficient, overall decreases mixing at higher altitudes and therewith atmospheric inversions are better preserved. A similar impact stems
from the last modification to the turbulence scheme we describe, decreasing the limiter on the minimum length scale, $l_{\infty}$, from 100 to 40 (section 2.3, Eq. (16)). The exact value of $l_{\infty}$ is highly uncertain, but also this parameter, active at higher altitudes, influences atmospheric inversion strengths. As demonstrated in the next sections, many of the improvements with cy40NEW arise from a more realistic representation of atmospheric inversions. In the next two sections we demonstrate the impact of the modifications on low clouds and low cloud base heights.





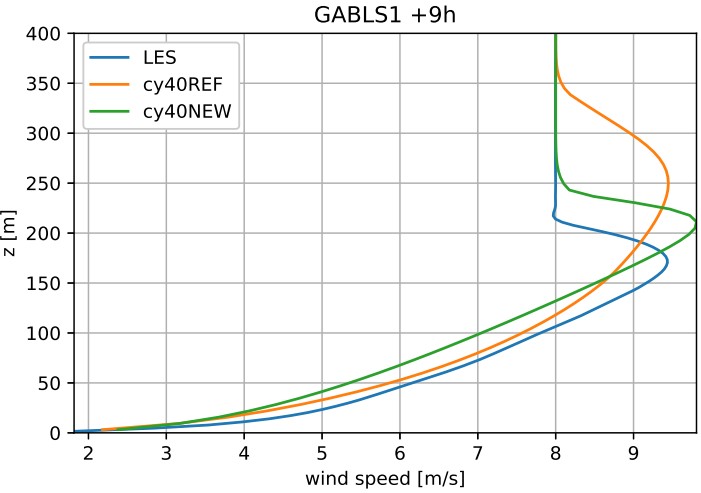

**Figure 13.** GABLS1 wind profile at the 9th simulation hour of LES model DALES (blue), cy40REF (orange) and cy40NEW (green). Note that results for GABLS1 with several LES models in (Beare and M.K. Macvean, 2006) reveal a spread in the height of the wind maximum, ranging from 175 to 200$m$. The latter height, and the corresponding LES profile in Beare and M.K. Macvean (2006) corresponds well with cy40NEW.

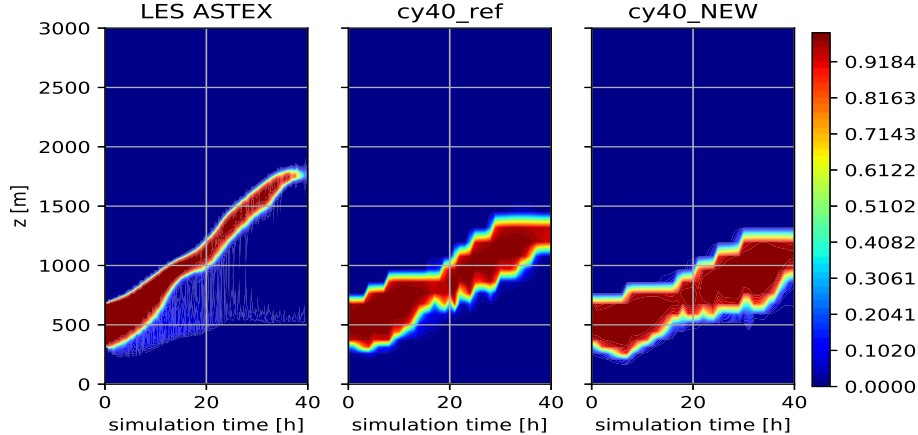

**Figure 14.** Cloud cover ASTEX case of LES (left panel), cy40REF (middle panel) and cy40NEW (right panel)





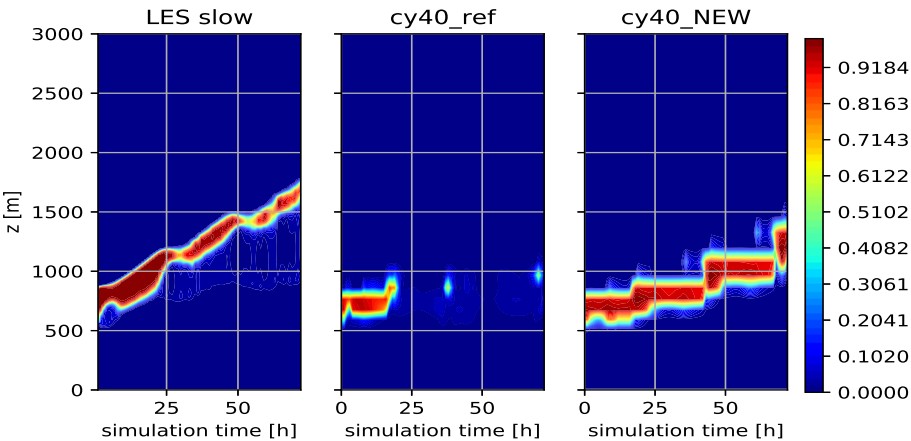

**Figure 15.** As Fig. 14 but for the Slow case

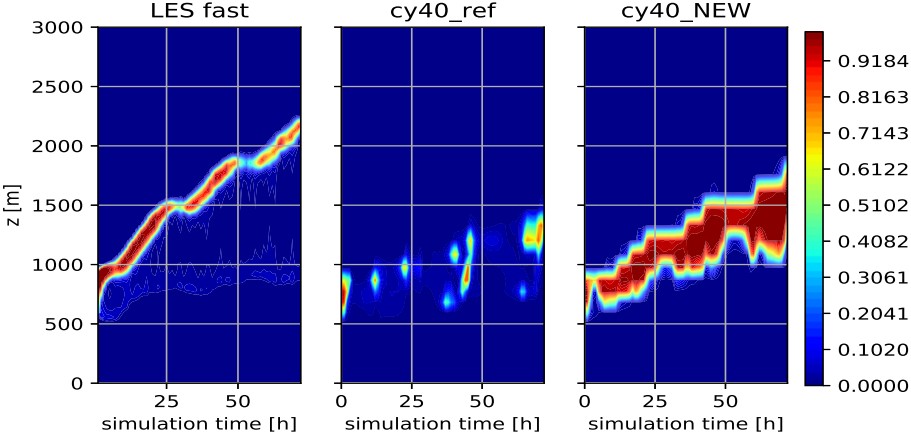

**Figure 16.** As Fig. 14 but for the Fast case

### 3.3 Stratocumulus to cumulus transition cases

Figures 14, 15 and 16 show the results of three strato-cumulus cases (see de Roode et al. (2016) and Neggers et al. (2017)). Whereas ASTEX is based on observations, the slow and fast case are composites, idealised cases. LES results are obtained with DALES (de Roode et al., 2016). SCM runs are performed with 80 vertical layers (slightly higher resolution than operational). SCM results for ASTEX are rather comparable although the new set-up shows a slightly thicker and less rising cloud layer, less in agreement with LES. Note that the lower vertical resolution in SCM's compared to LES, will usually lead to a more gradually rising cloud layer (Neggers et al., 2017). The slow and fast case (differentiated by the speed of the low-level cloud transition) however, illustrate the trouble of cy40REF to maintain a strato-cumulus layer, consistent with the strong underestimation of low



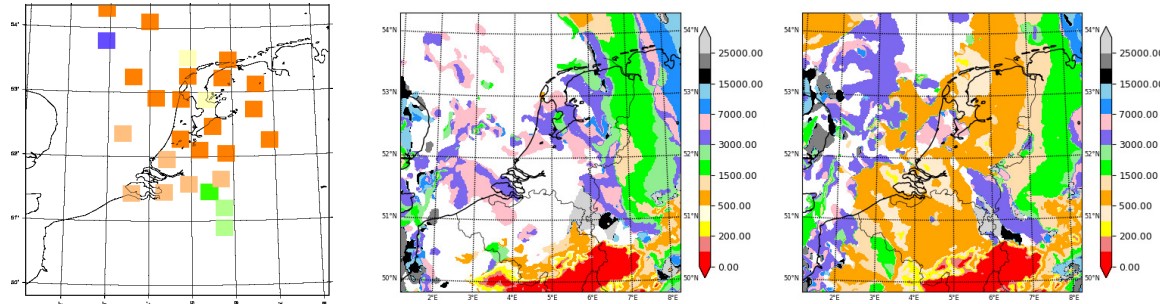

**Figure 17.** Cloud base height in feet at the 19th of December 2018 9:00UTC as measured at discrete observation site locations in The Netherlands and part of the North Sea (left panel), forecasted by cy40REF (middle panel) and cy40NEW (right panel). Note that white in the left panel means that there is no observation available whereas white spots in the middle and right panel mean no cloud base height detected because all model levels have a cloud fraction $< \frac{5}{8}$

clouds we see in operational practice. The improved results with the new set-up are related to the accumulated effect of several modifications. As a result of a more efficient moisture transport towards the inversion in combination with a decreased transport

through the inversion (better preservation of the inversion strength), more moisture is accumulated beneath the inversion, visible as a continuous and rising strato-cumulus layer in the cy40NEW runs (Figs. 15 and 16).

There is one specific difference between the model versions we need to mention concerning the slow case. In the results for this case only a moist updraft (see Fig. 4 right panel in B17) was invoked in cy40REF because the bulk difference in potential temperature between the surface and $700hPa$ exceeds the threshold of $20^oC$. The convective mixing with only

a moist updraft in cy40REF is unable to transport enough moisture to the inversion. Even when the temperature inversion between surface and $700hPa$ exceeds $20^oC$ it still seems legitimate to presume the existence of an ensemble of relatively weak, dry updrafts and stronger, moist updrafts. Moreover, rigid and rather arbitrary thresholds in parameterisations, like the above mentioned bulk temperature difference, should be avoided (Kähnert et al., 2021). Based on the considerations above, the stratocumulus regime with only a wet updraft is removed in cy40NEW.

**3.4 HARMONIE-AROME 3D model runs**

As mentioned in section 1, the most urgent problem in cy40REF concerns the large underestimation of low clouds and over-estimation of cloud base heights (i.e. the lowest model level where cloud fraction exceeds $\frac{5}{8}$). This model deficiency is most noticeable in winter time conditions. As a typical example we show 3D model results for the 19th of December 2018 in Fig. 17. The cy40REF run reveals a severe overestimation of cloud base height. Moreover, for large areas with observed low stratus,

cloud base height is not even detected due to too small cloud fractions (shown as white, background color). Key aspect of the large improvement with cy40NEW (Fig. 17 right panel) is again the better preservation of inversion strengths. Several modifications contribute to the improvement but most substantial is the influence of reduced $l_\infty$ (see Eq. (16)) and $c_h$ (Eq. (14)) as





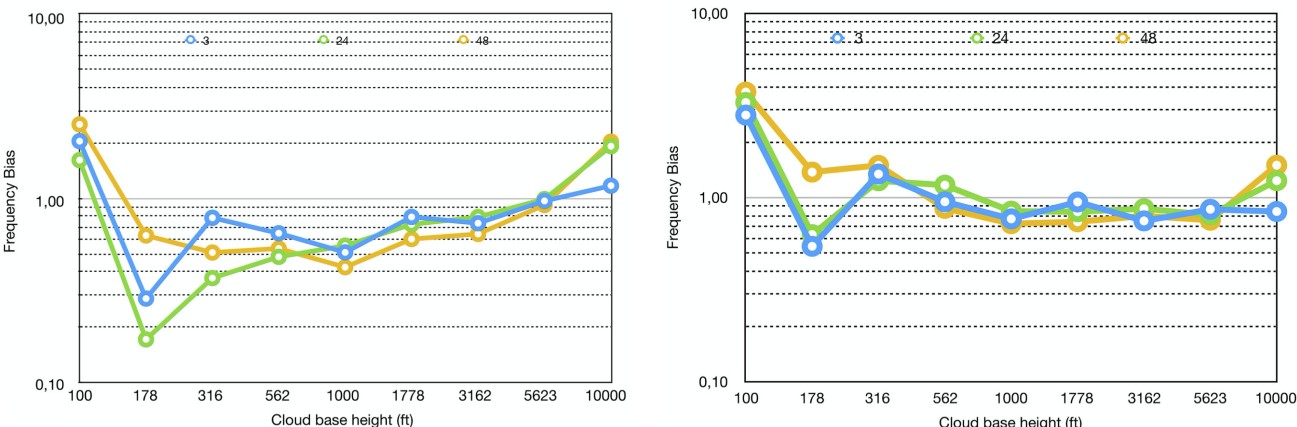

**Figure 18.** Frequency bias of the cloud base height for December 2018 with cy40REF (left panel) and cy40NEW (right panel). Blue, green and orange lines refer to resp. $+3h$, $+24h$, and $+48h$ forecasts

well as removal of the HARATU updates, increasing the downward mixing described in B17 (see also section 3.2). The large improvement on cloud base height is confirmed in longer term verification, illustrated by the frequency bias for December 2018

(Fig. (18)). Here frequency bias means the ratio between the forecasted and observed number of cloud base heights in a certain bin. Note the extreme underestimation of cloud bases around $178ft$; less than $20\%$ of the observed number of cases is actually predicted in $+24hr$ cy40REF forecasts. Over the complete range of low cloud base heights cy40NEW outperforms cy40REF, except for the lowest cloud base, associated with fog cases. However, in fog, other processes (concerning micro-physics and radiation) outside the scope of this study, turn out to have a large influence. Verification for other months confirm the substantial

improvement in low cloud base height climatology.

Apart from the impact on low clouds, the accumulation of moisture beneath atmospheric inversions also influences the triggering of resolved deep convection and the associated (heavy) precipitation. This is illustrated in Fig. 19 presenting a case on the $10th$ of September 2011 where deep convection was observed but its triggering was missed by cy40REF. The vertical atmospheric cross sections in Fig. 19 (third and fourth row) reveal that relative humidity just under the inversion of the

boundary layer accumulates more strongly in cy40NEW. This supports the model to start resolved upward motions as reflected in the increased boundary layer height near the local maximum in RH at the boundary layer top (fourth row, third column). As a result, only in cy40NEW deep, resolved convection and precipitation starts (blocky pattern in the upper right corner of the fourth row and column).

Semi-operational, daily runs of cy40REF and cy40NEW for more than a year in parallel, revealed several cases

where cy40NEW did forecast resolved precipitation that was also observed but was missed in cy40REF. Moreover, one year of fraction skill score verification of precipitation forecasts against calibrated radar data, demonstrated a significant improvement with cy40NEW (not shown).





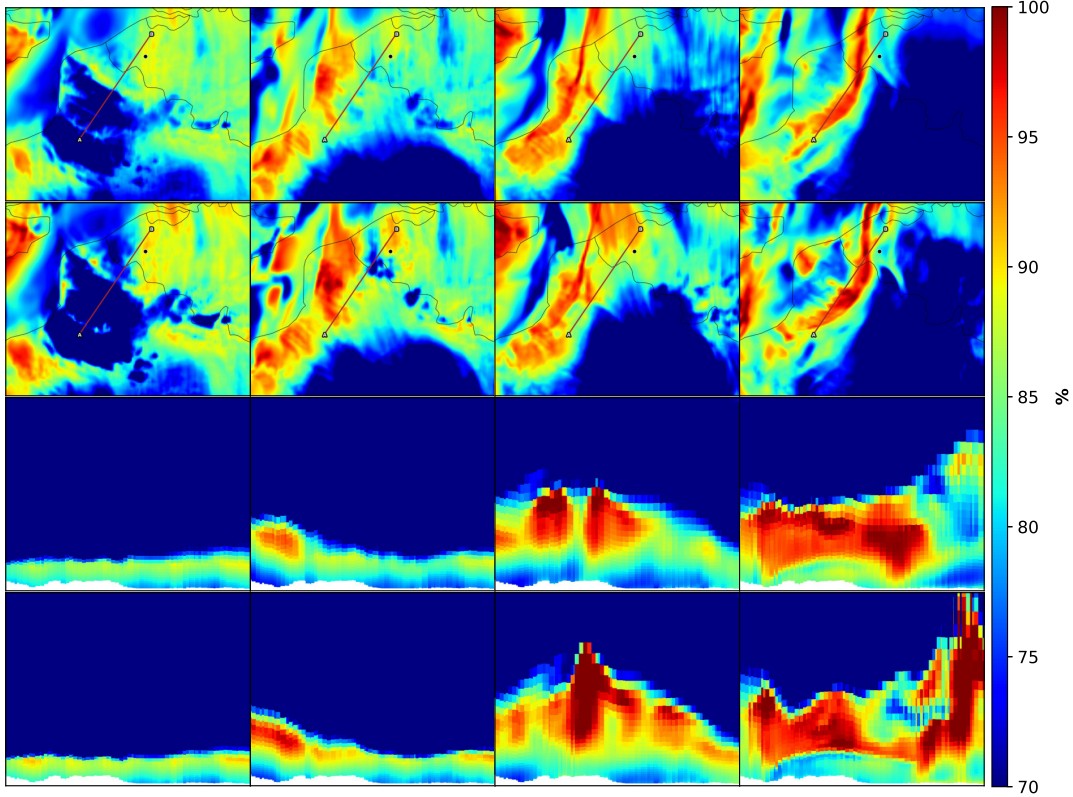

**Figure 19.** Relative humidity (RH) plots (red means high RH, blue low RH) for the 10th of September 2011. The four columns refer to hours 12, 14, 16 and 18 UTC. The first row(cy40REF) and second row (cy40NEW) show a map of RH at approximately $500m$ height that covers parts of Belgium and North-West France as well as a black line. Along this line a vertical atmospheric cross-section for the lowest $3km$ is shown in the third (cy40REF) and fourth (cy40NEW) row. In the cross-sections, the boundary layer can be recognised by relatively high RH values.

## 4 Conclusions and discussion

As discussed in e.g. Jakob (2010) or de Rooy et al. (2013), model development, in particular by means of improved param-
eterisation schemes, is a slow and sometimes frustrating process. A scientifically improved parameterisation could remove
a previous compensating model error and consequently cause an overall deterioration. In addition, together with increased
physical realism interactions between parameterisations become stronger. The considerations above advocate a more integral
approach to develop strongly connected parameterisation schemes together. Following such an approach, this paper describes
a comprehensive model update to the boundary-layer schemes. Because the involved parameterisations are all built on widely
applied frameworks, the here described modifications and the impact of certain parameters on different model aspects, is not
just specific to the HARMONIE-AROME model but applicable to many NWP and climate models. Moreover, this paper can



be an inspiration for further improvements and several suggestions for this are already provided. For example, amelioration of the variance in $s$ estimates by increasing the convection time scale, $\tau_{\mathrm{conv}}$ Eq. (31), or including a height dependency in the extra variance term, Eq. (38).

670 Apart from being a slow and tough process, model development is often a compromise between a scientific and a pragmatic approach. In this paper we have tried to provide an "honest" description of the development process so including the more pragmatic optimizations and mentioning not only the successes but also the remaining shortcomings and (over)simplifications in the parameterisations.

 The model update contains substantial modifications to the cloud, turbulence and convection schemes based on a
wide variety of argumentations. On one side of the spectrum are the more theoretically based modifications to the turbulence scheme (Monin-Obukhov similarity theory, following Baas et al. (2008) and Baas et al. (2017)) and the statistical cloud scheme (fundamental thermodynamics). On the other end of the spectrum, this paper illustrates that parameterisations contain uncertain parameters, with largely varying values suggested in literature, that at the same time have a substantial impact. To optimise these parameters we inevitably have to rely on examination of cases and longer term 3D runs. Finally, LES and SCM runs
conducted for a variety of intercomparison cases have been analyzed extensively and the outcomes are subsequently used as a basis for several modifications in all boundary layer schemes. As an example we mention the incorporation of the lateral mixing term from the prognostic mass flux vertical velocity variance equation as a source term in the TKE equation. This term is related to the energy cascade from large to smaller scales and particularly enhances the sub-cloud cloud layer transport improving the correspondence with LES results for shallow convection. An overview of all modifications is provided in Table
D1.

 The adjustments to the HARMONIE-AROME model described in this paper have a substantial impact on several aspects of the model performance. The most outstanding result is the improvement on low cloud and low cloud base height forecasts. Being one of the most urgent deficiencies of HARMONIE-AROME cycle 40, increasing the quality on this aspect was also the main goal of this study. The low cloud climatology changes from a severe underestimation in the reference
version to a well balanced model. Obviously, low clouds have a large impact on radiation and therewith on several model parameters. Moreover, they are crucial for aviation safety purposes. Taking a closer look at the consequences of the model updates reveals that the better preservation of atmospheric inversion strengths plays a key role. Not only the formation of low clouds, but also the triggering of deep resolved convection and the associated (heavy) precipitation is influenced by atmospheric inversion strength. With stronger inversions, more humidity is accumulated beneath the boundary layer top which supports the
development of meso-scale, resolved upward motions, ultimately leading to deep convection and rain showers.

 Verification based on more than one year of parallel model runs with cy40REF and cy40NEW firmly substantiates the significant improvement on low cloud and precipitation forecasts. Additionally, this long-term verification reveals the conservation of cy40REF's good performance on wind speed. All modifications have recently been incorporated in the default



configuration of HARMONIE-AROME cycle 43. Herewith, they will also become available in the HARMONIE-AROME
climate version (Belušić et al., 2020) with undoubtedly impact on e.g. precipitation extremes in future weather experiments.

An important spin-off of this project is the increased understanding in how parameter settings impact particular
model output and how they influence each other via underlying physical processes. With this insight we decided to use the
proportionality constant of the stable length scale, $c_{m,h}$ (14) and the minimum asymptotic length scale, $l_\infty$ (16) within a SPP
(stochastically perturbed parameterisations) EPS framework (Frogner et al., 2019). Verification reveals that these parameters
have the most benificial impact on spread/skill of all parameters investigated (pers. comm. Inger-Lise Frogner).

*Author contributions.* WdR contributed to all aspects of the paper including writing the original draft and revisions. PS, PB, GL and SdR
contributed to the conceptualization. PS and PB contributed to the formal analysis. PB, ST and BvtV contributed to the visualization. PS
contributed to the writing review editing. PS, PB, GL, SdR, HdV, EvM and JFM commented on the paper.

**Code availability**

The ALADIN and HIRLAM consortia cooperate on the development of a shared system of model codes. The HARMONIE-
AROME model configuration forms part of this shared ALADIN–HIRLAM system. According to the ALADIN–HIRLAM col-
laboration agreement, all members of the ALADIN and HIRLAM consortia are allowed to license the shared ALADIN–HIRLAM
codes to non-anonymous requests within their home country for non-commercial research. Access to the full HARMONIE-
AROME codes can be obtained by contacting one of the member institutes of the HIRLAM consortium (see links at:
http://www.hirlam.org/index.php/hirlam-programme-53) and is subject to signing a standardized ALADIN–HIRLAM licence
agreement (http://www.hirlam.org/index.php/hirlam-programme-53/access-to-the-models).

The code of all routines involved in the modifications described in this paper, together with the corresponding original
routines are available as electronic supplement and can be obtained upon request. The supplement retains the directory struc-
ture as in the full Harmonie-Arome model. Directory src/arpifs/phys_dym contains four modified routines: apl_arome.F90,
vdfexcuhl.F90, vdfhghtnhl.F90 and vdfparcelhl.F90 that involve changes to, respectively the cloud scheme, the turbulence
scheme, the convection and turbulence scheme, and finally the convection scheme. Corresponding original routines are al-
ways indicated by the extension _ori. Directory mpa/micro/internals includes condensation.F90 with modifications to the
cloud scheme. Finally, directory mpa/turb/internals contains five routines with modifications to the cloud scheme: com-
pute_function_thermo_mf.F90, compute_mf_cloud_stat.F90, ini_cturb.F90, turb.F90 and turb_ver_thermo_corr.F90. In the
same directory, two routines include modifications related to the turbulence scheme: turb_ver_dyn_flux.F90 and
turb_ver_thermo_flux.F90. With reference to this paper, all routines in the supplement file can be freely used e.g. in other
software.





**Competing interests**

The authors declare that they have no conflict of interest.

**Acknowledgements**

This work has been done within the KNMI multi-annual strategic research (MSO) project CRIME (Cloud Representation IM-
provement and Evaluation in HARMONIE-AROME) and the Norwegian Research Council project no. 280573, 'Advanced
models and weather prediction in the Arctic: enhanced capacity from observations and polar process representations (ALERT-
NESS)'. Support of Emiel van de Plas with python is appreciated

**Appendix A**

**A1    Derivation of the variance in $s$**

Here we provide a step by step derivation of the variance in $s$.

Suppose we know the PDF that describes subgrid variability of $\theta_l$ and $q_t$ in a grid box of an atmospheric model. Then
the resulting cloud cover, $a_c$ and liquid water content (similarly for ice water content) can be written as:

$$a_c = \int_{-\infty}^{\infty} H(q_t - q_s) P(\theta_l, q_t)\, \mathrm{d}q_t \mathrm{d}\theta_l$$

$$q_l = \int_{-\infty}^{\infty} (q_t - q_s) H(q_t - q_s) P(\theta_l, q_t)\, \mathrm{d}q_t \mathrm{d}\theta_l \tag{A1}$$

where $q_s$ is the saturation specific humidity and $H$ denotes the Heaviside function ($H(x) = 0$ for $x < 0$ and $H(x) = 1$ for
$x > 0$) which probes that part of the integrand that is over-saturated. Because we only have to consider $q_t - q_s > 0$, the distance
to the saturation curve $s$ can be defined as

$$s \equiv \overline{s} + s' = q_t - q_s(p, T) = q_l \quad for \quad s > 0 \tag{A2}$$

where $\overline{s}$ is the (grid box) average of $s$, primes denote excursions from the mean and $q_s$ is a function of pressure, $p$, and
temperature $T$. Using a Taylor expansion around $\overline{T_l}$, the saturation specific humidity at $T$ can be written as:

$$q_s(T) \simeq \overline{q}_{sl} + \overline{q}_{sl,T}(T - \overline{T}_l) \tag{A3}$$





with the usual abbreviations:

$$\overline{q}_{\mathrm{sl}} = q_{\mathrm{s}}(\overline{T}_1), \quad \overline{q}_{\mathrm{sl,T}} = \frac{\partial q_{\mathrm{s}}(\overline{T}_1)}{\partial T} \tag{A4}$$

using the definition of the liquid water temperature:

$$T_{\mathrm{l}} \equiv T - \frac{L}{c_{\mathrm{p}}} q_{\mathrm{l}}, \tag{A5}$$

where $L$ is the latent heat of vaporization and $c_{\mathrm{p}}$ the heat capacity of dry air at constant pressure. Equation (A3) can be rewritten

as:

$$q_{\mathrm{s}}(T) \simeq \overline{q}_{\mathrm{sl}} + \overline{q}_{\mathrm{sl,T}}(T_{\mathrm{l}} + \frac{L}{c_{\mathrm{p}}} q_{\mathrm{l}} - \overline{T}_1) = \overline{q}_{\mathrm{sl}} + \overline{q}_{\mathrm{sl,T}}(\pi\theta'_{\mathrm{l}} + \frac{L}{c_{\mathrm{p}}} H(s)s) \tag{A6}$$

where we have applied (A2) and the Exner function, $\pi = \left(\frac{p}{p_0}\right)^{\frac{R_{\mathrm{d}}}{c_{\mathrm{p}}}} = \frac{T}{\theta}$ with $R_{\mathrm{d}}$ is the gas constant of dry air and $p_0$ is a reference surface pressure. Equation (A6) substituted in Eq. (A2) leads to:

$$s = \overline{q}_{\mathrm{t}} + q'_{\mathrm{t}} - \overline{q}_{\mathrm{sl}} - q_{\mathrm{sl,T}}\pi\theta'_{\mathrm{l}} - q_{\mathrm{sl,T}}\frac{L}{c_{\mathrm{p}}} H(s)s \tag{A7}$$

As mentioned before, we only consider $s > 0$, so $H(s) = 1$. Writing $s$ explicitly in (A7) leads to:

$$s = \alpha[q'_{\mathrm{t}} - \beta\theta'_{\mathrm{l}} + (\overline{q}_{\mathrm{t}} - \overline{q}_{\mathrm{sl}})] \tag{A8}$$

with $\alpha$ and $\beta$ defined in Eq. (22). To determine $s'$ we follow a similar derivation as shown above but now for $\overline{s}$.

$$\overline{s} = \overline{q}_{\mathrm{t}} - \overline{q}_{\mathrm{s}}(\overline{T}) \tag{A9}$$

$$\overline{q}_{\mathrm{s}}(\overline{T}) \simeq \overline{q}_{\mathrm{sl}} + \overline{q}_{\mathrm{sl,T}}(\overline{T} - \overline{T}_1) \tag{A10}$$

with $(\overline{T} - \overline{T}_1) = \frac{L}{c_{\mathrm{p}}}\overline{q}_{\mathrm{l}} = \frac{L}{c_{\mathrm{p}}}\overline{s}$ substituted in (A9), $\overline{s}$ reads:

$$\overline{s} = \alpha(\overline{q}_{\mathrm{t}} - \overline{q}_{\mathrm{sl}}) \tag{A11}$$

Using eqs. (A8) and (A11) we can write $s'$ as:

$$s' = s - \overline{s} = \alpha q'_{\mathrm{t}} - \alpha\beta\theta'_{\mathrm{l}} \tag{A12}$$

and the variance of $s$ as:

$$\sigma_{\mathrm{s}}^2 = \overline{s'^2} = \alpha^2\overline{q'^2_{\mathrm{t}}} - 2\alpha^2\beta\overline{q'_{\mathrm{t}}\theta'_{\mathrm{l}}} + \alpha^2\beta^2\overline{\theta'^2_{\mathrm{l}}} \tag{A13}$$





## A2 Summary of the differences between the cy40REF and cy40NEW cloud scheme

Here we present an overview of the differences between the cy40REF and cy40NEW cloud scheme. Firstly, an important difference concerns the formulation of the thermodynamic coefficients $\alpha$ and $\beta$ in the expression for the variance in $s$ (21).

The definitions and derivation in cy40NEW can be found in the previous appendix. In cy40REF coefficient $\alpha$ is formulated as (22) except for a factor $0.5$ (see Tudor and Mallardel (2004)). Coefficient $\beta$ in cy40REF is combined with $\alpha$ in one variable in a complex expression, described in Tudor and Mallardel (2004) but without a derivation or reference. The values and typical atmospheric shape of the profile of $\alpha$ in the original code are wrong as they deviate substantially from (22) (not shown). Furthermore, in cy40REF it is assumed that $l_\varepsilon$ equals $l_\mathrm{m}$ (Eq. (30)), whereas in the new configuration we take $l_\varepsilon$ consistent

with its formulation in the turbulence scheme (see Eq. (27)). Pre-factor $c_\mathrm{ab}$ in Eq. (26) was $1$ in cy40REF but changed to $0.139$, this time conform literature (Redelsperger and Sommeria, 1981). In contrast with the reference code, the new set-up of the cloud scheme includes the covariance term of the contribution from convection, i.e. Eq. (31) with $a = \theta_\mathrm{l}$ and $b = q_\mathrm{t}$. Finally, prefactor $2$ of the variance contribution from convection (see e.g. Eq. (32)) was erroneously applied twice in cy40REF and removed in cy40NEW.

## Appendix B: Modifications in the convection scheme

To estimate the contribution from organised (updraft) transport, in a model represented by the convection scheme, to the total turbulent transport, LES data in the cloud layer is conditionally sampled. Different sampling methods exist (see Siebesma and Cuijpers (1995)) like cloudy updraft sampling, i.e. selecting LES gridboxes with $w_\mathrm{u} > 0$ and $q_\mathrm{l} > 0$, and core sampling with the additional requirement of positive buoyancy. Cloudy updraft sampling is probably most suitable to be compared with

convective transport of a mass flux scheme because it includes the negatively buoyant, decelerating part of the updraft, just as in the parameterisation.

Figure B1 shows convective humidity transport according to LES (cloudy updraft sampling) and HARMONIE-AROME 1D with the cy40REF and cy40NEW configuration. Plots of heat transport are not shown as they reveal a similar behaviour. The plotted HARMONIE-AROME values are the sum of dry and moist updraft transport whereas the sampling

method applied on the 3D fields of LES will only produce estimates of convective transport in the cloud layer. To increase statistical significance, the model mass flux transport is obtained as hourly mean around validation time. From LES only instantaneous hourly 3D fields are available. However, as LES convective transport is the mean of 100 Harmonie-sized domains, it can be considered as an average over many realizations.

Fig. B1 shows that during the main part of the convective period, both model versions underestimate convective

transport in comparison with LES. Only during the last convective hours, fluxes are comparable whereas at $+12h$ convection finally starts to collapse. The latter hour is highly dynamical and a slightly different (e.g. shorter) averaging time already



**Figure B1.** Kinematic convective transport $\left[\frac{m}{s}\right]$ during all convective hours of the ARM case, corresponding to simulation hours $+4$ to $+12$ hours. Plotted is the mass flux (MF) transport by the convection scheme (orange=cy40REF and green is cy40NEW) and the estimated (cloudy updraft sampling) convective transport by the LES (blue). Note that the x-axis scale is not constant and equal to the scale of the corresponding plots in Fig. 4





has a large impact on the diagnosed flux profiles. Hence, $+12h$ results should be interpreted with care. Figure B1 further demonstrates that the new configuration increases convective transport, generally resulting in a better resemblance with LES. Several modifications in the convection scheme have contributed to this increase in mass flux transport. All modifications to
the convection scheme, including their impact, are described below.

Firstly, we changed $c_b$ in the mass flux closure (12) from $0.03$ (Grant, 2001) to $0.035$ (Brown et al., 2002), see section 2.2. Another contribution stems from the formulation of $\varepsilon$ at $z = z_{lcl}$ (Eq. (10), section 2.2.1). In the original expression, entrainment at cloud base (or inversion height) is inversely proportional to the inversion height. With a typically increasing inversion height during the convective period this formulation will result in relatively high entrainment rates, and therewith less
effective mass flux transport in the early stages of convection. However, during this period the convective transport is underestimated (see Fig. (B1)). Therefore, we fixed moist updraft entrainment values at cloud base at $0.002$, roughly in agreement with LES (de Rooy et al. (2013), Fig. 6 and Siebesma et al. (2003)). Another aspect of the entrainment formulations in cy40REF are the quite large values near the surface due to the first term on the RHS in Eqs. (9) and (10). Apart from unwanted dependence on vertical resolution of the model this will also result in a weak dependence of updraft excess values on surface fluxes. By
adding $a_1 = 40m$ to the entrainment formulations, similar to (Soares et al., 2004), dependence on surface fluxes gets stronger, causing increased convective transport during hours with large surface fluxes (see Fig. 3 in Brown et al. (2002)). Finally, $a_2$ in Eq. (9) is reduced from $40m$ to $1m$ to increase entrainment values when the dry updraft approaches its termination height. Herewith, deposition of humidity in a too thin layer just below the inversion is prevented, which contributes to the too high humidity and cloud cover around cloud base in cy40REF (see section 3.1.2).

Finally, Fig. B1 reveals a strong decrease in mass flux transport around inversion which is related to the termination height of the dry updraft and the associated strong decrease of convective transport. However, as we demonstrate in section 3.1.1, this decrease in convective transport is largely balanced by the diffusive transport leading to a rather smooth total turbulent transport profile (Fig. 4).

**Appendix C: Decomposition of the turbulent transport**

Following Siebesma and Cuijpers (1995), total turbulent transport can be written as a sum of large-scale organised and small-scale sub-plume and environmental transport. Fig. C1 presents typical profiles during the ARM case of such a decomposition of total turbulent transport. The role of environmental turbulence in Fig. C1 is remarkable. In the lower half of the cloud layer the negative contribution of environmental turbulence is roughly balanced by positive sub-plume turbulence. However, in the upper part of the cloud layer a large negative contribution of environmental turbulence dominates and counteracts organised updraft
transport. Consequently, the underestimation and too shallow organised convective transport by the parameterisation (Fig. B1) is not translated in an underestimation of total turbulent transport (Fig. 4). Note that in Siebesma and Cuijpers (1995) Fig. 7 for the BOMEX steady state shallow convection case, environmental turbulence is always positive. Their figure is produced





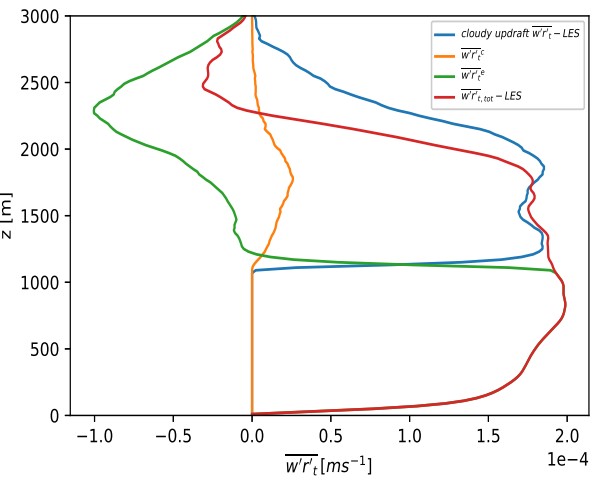

**Figure C1.** Decomposition of the turbulent fluxes for the ARM case, 9th simulation hour. Plotted are LES cloudy updraft flux (blue), small-scale sub-plume transport (orange), small-scale environmental transport (green), and total transport (red).

applying cloud core sampling. However, repeating the decomposition experiments with different sampling methods lead to the same qualitative picture.

To investigate the relatively large contribution from environmental turbulence, the turbulent transport is decomposed further in three parts: cloudy updraft, cloudy downdraft and environment (see Siebesma and Cuijpers (1995)). As a result we now distinguish 6 different turbulent fluxes contributing to the total turbulent transport of moisture (Fig. C2). Figure C2 reveals that less than half of the negative turbulent transport is caused by organised downdrafts whereas the majority is caused by environmental turbulence outside cloudy up- and downdrafts. To visualise the downward transport, a horizontal cross section

is taken at the height of maximum downward turbulent moisture transport (Fig. C3). The largest downward transport (dark blue color) is observed in two sub-domains indicated by black squares and seems to be connected to strong upward transport. However, the two sub-domains reveal a different behaviour (Figs. C3 middle and right panel). Whereas the right sub-domain resembles the classical view with downward transport in the cloud (downdrafts), the left sub-domain shows downward transport primarily outside the cloud (indicated by the red $q_l = 0$ line), possibly the remains of a large active updraft. Here, a substantial

part of downward transport is associated with downdrafts containing relatively high humidity values but no liquid water. Possibly, these downdrafts are related to the subsiding shells as discussed by Heus et al. (2009).

Finally, Fig. C3a illustrates that LES runs for the ARM case at a smaller domain could easily miss rarely occurring large convective events that give rise to substantial downward transport. As a result, investigations on smaller domain LES could lead to different conclusions about the relative importance of the decomposed fluxes to the total turbulent transport.



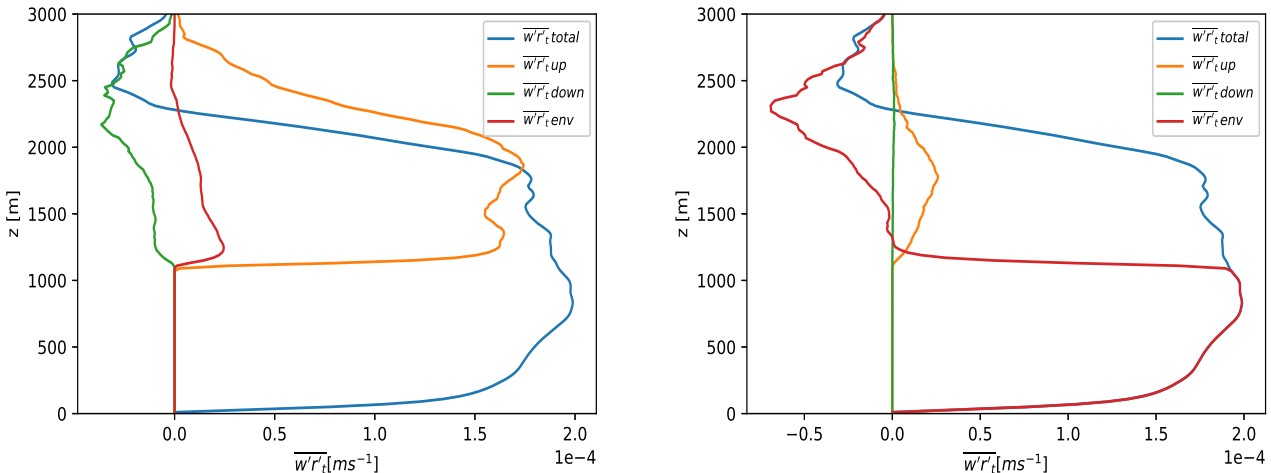

**Figure C2.** ARM case, 9th simulation hour. The left panel shows organised fluxes, distinguishing updrafts (orange), downdrafts (green) and environment (red) as well as the total turbulent transport (blue). The right panel shows the small-scale turbulent fluxes using similar color coding as in the left panel

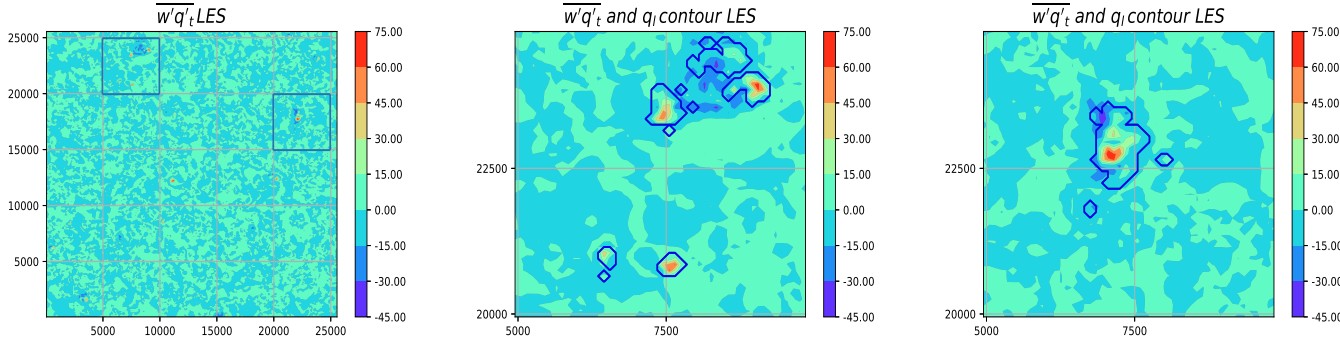

**Figure C3.** ARM case, 9th simulation hour, cross section of the kinematic turbulent moisture transport at 2310m height (with $q_t$ in $\frac{g}{kg}$). Blue and yellow/red colors refer to resp. downward and upward transport. The x and y-axis number the LES grid points (with the LES resolution of $100m$; the gray grid lines illustrate the size of a HARMONIE-AROME grid box). The left panel presents the full LES domain whereas the middle and right panel show resp. the left and right sub-domains as shown by the black squares in the left panel. The red line defines the cloudy border, i.e. $q_l = 0$.





**Appendix D: Overview of the modifications**





| description | cy40REF | cy40NEW | main impact/argumentation |
|---|---|---|---|
| **Shallow convection scheme** | | | |
| separate regime for strato-cumulus? Section 2.2, Table 1 | yes: $a_{dry} = 0m$ | no: as in shallow cumulus i.e. $a_{dry} = 0.07m$ | improvement in strato-cumulus cases and remove arbitrary threshold |
| entrainment Eqs. (9), (10), (11) | $a_1$ not present | $a_1 = 40m$ | reduce dependence on vertical resolution and increase dependence on surface fluxes |
| | $a_2 = 40m$ | $a_2 = 1m$ | prevent humidity deposition in a too thin layer below inversion |
| | $\varepsilon_{lcl} = \frac{1.65}{z_{z_{lcl}}} m^{-1}$ | $\varepsilon_{z_{lcl}} = 0.002 m^{-1}$ (Siebesma et al., 2003), (de Rooy et al., 2013) | increase mass flux transport in the early stages of convection (conform LES) |
| mass flux closure Eq. (12) | $c_b = 0.03$ (Grant, 2001) | $c_b = 0.035$ (Brown et al., 2002) | increase mas flux (conform LES) |
| **Turbulence scheme** | | | |
| proportionality constant of stable length scale for heat, Eq. (14) | $c_h = 0.15$ | $c_h = 0.11$ | mixing for neutral to moderately stable conditions tuned against MO-theory |
| power of the inverse interpolation between length scales, Eq. (15) | $p = 1$ | $p = 2$ | as above |
| aymptotic free atmospheric length scale, Eq. (16) | $l_\infty = 100m$ | $l_\infty = 40m$ | stronger atmospheric inversions and better preservation stratus clouds |
| turbulent diffusion link to convection, Eq. (18) | $50 \cdot \mathcal{M}_u$ | energy cascade $W_{casc}$ | improved turbulent transport (sub-cloud to cloud transport) conform LES |
| enhanced downward mixing in storm situations | included see Bengtsson et al. (2017) | removed | removed due to retuned $c_h$ and $p$ |
| **Cloud scheme** | | | |
| thermodynamic coefficients $\alpha$ and $\beta$, Eqs. (21), (22) | Tudor and Mallardel (2004) (bug) | see appendix A1 | bug removal |
| dissipation length scale, Eq. (27) | $l_\epsilon = l_m$ | $l_\epsilon = c_0^2 l_m$ (LH04) | now consistency between turbulence and cloud scheme |
| dissipation term constant, Eq. (26) | $c_{ab} = 1$ | $c_{ab} = 0.139$ | now conform literature (Redelsperger and Sommeria, 1981) |
| covariance term in the contribution of convection to $s'^2$, Eq. (31) | not included | included | improves the shape of variance profile |
| convective contribution to the variance, see Eqs. (31), (32) | erroneous extra factor 2 | - | bug removal |

**Table D1.**



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
