# Peer review of "Model development in practice: A comprehensive update to the boundary layer schemes in HARMONIE-AROME cycle 40"

_Geoscientific Model Development, 2021_

## Referee Comment (RC1)

**Manuscript number**: gmd-2021-184

**Title**: Model development in practice: A comprehensive update to the boundary layer schemes in HARMONIE-AROME cycle 40

**By**: de Rooy and Coauthors

**General comments and recommendations**

The authors provide a detailed description of a set of updates implemented simultaneously in three parametrizations – boundary layer turbulence, shallow convection and cloud schemes – in the HARMONIE-AROME model. The main goal of those updates was to improve the representation of low clouds in that model, which is indeed a challenge faced by many modellers.

I think that the manuscript is well-written, the description of the parametrizations is sufficiently clear, the methods employed to evaluate the impact of model changes are appropriate, the choice of figures and tables is adequate, and the additional material in the appendices is useful. I also think that the contents of the manuscript are of particular interest to the modelling community.

Therefore I believe that the manuscript could be published, once the minor points below are addressed.

**Minor points**

- I understand that the main goal was to improve the quality of low cloud forecasts. I would expect that a significant change in clouds would impact near-surface temperatures and moisture, but I did not find a comment about it in the manuscript (only a few comments about the impact on precipitation and winds, in section 4). I suggest that the authors consider adding some comments on that.
- Section 2.1, line 119: described -> described
- Section 2.2, line 133: epsilon_k : I suggest that you use the same symbol/Greek letter used in eq. 6
- Section 2.2, eq. 7: Should it be overline(theta_v) in the denominator?
- Section 2.2, line 137: "updrafts are initialised at the lowest model level" … does this imply a un-realistic dependence on vertical resolution?
- Section 2.2, line 139: "variance estimated from the surface fluxes"… are you referring to turbulent fluxes? Please clarify.
- Section 2.2.1, line 191: "very rapidly" …. how rapidly? Please clarify.
- Section 2.2.1, eq. 9: Please state the z-range of validity (e.g. is it valid for z <=z_i,dry , then zero elsewhere?)
- Section 2.3, line 305: lineair -> linear
- Section 2.4, eq. 32: Should it be l_h instead of l_{m,h}? Since this equation refers to theta_l (not momentum)?

- Section 3.1, line 474: I suggest to replace "cherry picking" by a less informal expression.
- Fig. 12: Please explain the black dots.
- Fig. 19: If possible, I suggest that a vertical scale be added to the vertical axis of the figures in the 3$^{rd}$ and 4$^{th}$ row (the vertical cross sections).
- Line 653: I'm not sure I understand "blocky pattern", so I suggest a clarification, or instead a less informal expression.

---

## Referee Comment (RC2)

Review of 'Model development in practice: A comprehensive update to the boundary layer schemes in HARMONIE-AROME cycle 40' by de Rooy et al.

gmd-2021-184

This paper presents a detailed description of a package of changes to the representation of physical processes in the HARMONIE-AROME NWP model. The authors do, as they suggest, provide an "honest" description of their development process that is also informative and likely to be of widespread interest (beyond users of that particular model). Extensive analysis of the impact of the changes is given from idealised single-column model simulations to objective verification in NWP trials.

Overall the paper is well written and strikes a good balance in terms of the level of detail, given the breadth of schemes being altered in the package, and does a good job of explaining the motivation for the changes, be they from theorectical considerations, detailed analysis of LES or pragmatic changes to improve performance. I have got quite a few detailed questions and comments below, the most significant of which concerns a lack of clarity in the logic underlying the various updraughts used - I recommend this requires a careful review of the text and perhaps the addition of a flow diagram. A more general comment concerns the lack of attention given to momentum mixing, despite concern over wind speed forecasts prior to cycle 40. In particular, no mention is made of whether there is any momentum mixing by the massflux schemes. At least some comment on this aspect is required.

Further details and more minor comments are given below.

1. line 43, "cloud, turbulence and cloud scheme": two cloud schemes, one of which should be convection!

2. lines 138 and 202: good that you ackowlegde this inherent dependency on the height of the lowest model level in the entrainment rate but the dependence of the initial parcel properties feels like a much stronger one that you don't seem to worry about. Building in an unnecessary resolution dependence always seems like a bad idea. A physical height, such as the top of the surface layer would seem much better, given you are adding perturbations scaled by the surface fluxes which will quantify the near-surface gradients well enough

3. line 141: are the updraught area fractions really constant with height or does Table 1 show the initial fractions? Later on you are explicit that this is the case for the dry updraught (line 236) but for the moist updraught, given you have separate $w$ and $M$ profiles at least in the cloud layer, then that would imply you have an updraught area that varies? I can't see any use made of it, though, such as in the cloud scheme (see additional comment below)?

4. line 143: I don't understand the need for an a priori diagnosis of regime. Why not initialise both dry and moist plumes at the surface, calculate their evolution with height and from that diagnose if clouds are possible (based on the moist parcel reaching its LCL)? Is it just to save cpu time?

5. section 2.1.1: I'm confused by this description of the different parcels. Perhaps a logic flow diagram would help but several questions arise. (i) you say the test parcel is used to determine an estimate of the inversion height, so how can the moist updraught LCL come into this (line 179)? (ii) in line 190 you say "this iteration process converges very rapidly" but don't say anything about what steps are iterated and how the iteration is monitored or convergence measured. (iii) can you really be confident that if the test updraught doesn't reach its LCL then the moist parcel will not reach its own? Given the different formulations this doesn't seem certain. (iv) around line 200 you suggest the dry parcel cannot reach $z_{i,dry} + a_2$ but isn't $z_{i,dry}$ just an estimate from the rather different test parcel (as you can't know the inversion height for the dry parcel without knowing $\epsilon_{dry}$), so what stops the dry parcel below $z_{i,dry} + a_2$ in practise? (v) in (10) is $z_{lcl}$ the LCL of the test parcel as described in line 183? If so it would be good to make that definition of $z_{lcl}$ explicit. But I'm concerned that you specify a change in the moist updraught's entrainment rate at $z_{lcl}$ even though its LCL is likely to be different, not least because of the different fractional areas and sub-cloud entrainment rates. Does this not matter that the moist parcel's LCL may not conincide with where you change its entrainment rate?

6. line 216:, "deeper boundary layers will contain larger updrafts with relatively small entrainment values": the references supplied don't actually show that this dependence is wrong - perhaps the specific formulation in the REF scheme is not good for the ARM case, or do you have other concerns?

7. line 217, "Eq. (10) shows an inverse correlation between updraft vertical velocity and entrainment magnitude": it doesn't actually, but I think you are motivating its shape from expected vertical velocities?

8. line 269, "from there mass flux decreases linearly to 0 at cloud layer top": this sounds like a rather crude assumption and no massflux profiles are shown to illustrate its success or otherwise. For example, I might expect the massflux decrease to be constrained much closer to a sharper inversion, eg one maintained by radiative cooling of a stratiform layer at the cumulus top (as in the transition SCM cases shown in 3.3) than in the ARM case where shallow cumulus detrainment into a stable stratification is probably what determines the inversion thickness

9. line 357, "presume a Gaussian PDF": this seems like a big assumption, especially for cumulus clouds where the pdf is quite likely to be skewed?

10. Fig 4: I think you could usefully and safely (ie without cluttering the plot) add the dry and moist massflux scheme components (eg as dotted and dashed lines) for the cy40 simulations, so we can see the relative contributions. I think that would give valuable insight into the workings of the overall parametrization

11. line 460: you give the horizontal resolution but not vertical. Including the height of the bottom grid-level is also clearly important for the parcel initial properties.

12. line 494, "inclusion of the energy cascade": this does clearly improve the ultimate fluxes but it would be good to know it also improved the TKE profiles, which are not currently shown, and so improving performance for the right reason. For example, you also speculate (line 517) that "a plausible explanation for the presence of diffusive transport...are (dry) updrafts terminating around the inversion height" but

could you not be underestimating the transport by those dry updraughts themselves (again, showing the break down in Fig 4 would be instructive here)?

13. line 550, "underestimation of low values of cloud fraction in the upper part of the cloud layer": it is hard to work out the contour interval from the colourbar but the LES looks only to have a cloud fraction of a few percent, which could be similar to your moist updraught area. Are you not missing this (highly skewed, see above) contribution to cloudiness?

14. line 579: please could you give more detail of how the SCM is forced from RACMO (horizontal advection, surface fluxes or interactive land surface?), when are the forecasts initialised and how long are they for?

15. line 601: as noted in Beare and MacVean for GABLS, many NWP centres find they have to bias their turbulence scheme away from LES in order to improve objective verification of the forecasts. Is this not the case for you and, if not, do you have any insights as to why?!

16. line 623: I don't see why you would invoke a different set-up in this case only, especially for the REF scheme? Isn't the purpose to illustrate operational performance in simple test cases? It is then not clear, when you say (line 628) "Based on the considerations above, the stratocumulus regime with only a wet updraft is removed in cy40NEW", whether that is just in the SCM simulation or is this the motivation for this change in all tests?

17. line 636, "Key aspect of the large improvement with cy40NEW is again the better preservation of inversion strengths": this statement would be much stronger if backed up by a sample profile or cross section showing this sharper inversion.

18. line 638, "removal of the HARATU updates": these were reported (line 278) as being needed to alleviate problems with wind speeds so how do those look in cy40NEW? You do (finally, line 698) say the performance is maintained but in what metric (diurnal cycle of wind speed bias would be a good one to show)?

19. Table D1, Shallow convection scheme, cy40REF: I suspect the formula for $\epsilon_{lcl}$ has too many layers of subscript in $z_{z_{lcl}}$?

---

## Author Response (AR1)

Reply to reviewer 1 of gmd-2021-184

We thank reviewer 1 for the positive words and the useful comments and recommendations, which helped to substantially improve the paper. Below we describe how we addressed your specific points. Hopefully, you can now accept our paper for publication.

Kind regards

Wim de Rooy and co-authors

*- I understand that the main goal was to improve the quality of low cloud forecasts. I would expect that a significant change in clouds would impact near-surface temperatures and moisture, but I did not find a comment about it in the manuscript (only a few comments about the impact on precipitation and winds, in section 4). I suggest that the authors consider adding some comments on that.*

Indeed, near-surface variables are influenced by the model update. We now describe the main (small) overall impact on 2m temperature and humidity as well as 10m wind speed. In general the wind speed itself increases somewhat due to the modifications but (amplitude of) the diurnal cycle is comparable. Small improvements as well as small deteriorations in the near surface output variables can be found depending on the choice of the month or domain. Therefore, we think that adding near surface verification plots would require quite some additional explanation and would distract from the main subject and impact. We now also mention that near-surface parameters are strongly influenced by surface processes (not involved in this study) and representation mismatch between grid box and observation site conditions.

*- Section 2.1, line 119: described -> described*

Done

*- Section 2.2, line 133: epsilon_k : I suggest that you use the same symbol/Greek letter used in eq. 6*

Done

*- Section 2.2, eq. 7: Should it be overline(theta_v) in the denominator?*

You are right! Done.

*- Section 2.2, line 137: "updrafts are initialised at the lowest model level" … does this imply a unrealistic dependence on vertical resolution?*

It seems plausible that a lower resolution, that is a higher first model level, results in a larger excess value of the updraft (compared to the environment). This because of the decrease of grid box temperature and humidity values with height. However, due to the $z^{-1}$ dependence of the entrainment formulation the excess values decreases with a similar amount as the background. This is explained in detail in Appendix A of Siebesma et al. JAS, 2007 (A Combined Eddy-Diffusivity Mass-Flux Approach for the Convective Boundary Layer). As a result the initialization is rather independent on vertical resolution. This is now mentioned in the text.

*- Section 2.2, line 139: "variance estimated from the surface fluxes"… are you referring to turbulent fluxes? Please clarify.*

We now mention **turbulent** surface fluxes following the standard surface layer scaling of Wyngaard et al. 1971.

*- Section 2.2.1, line 191: "very rapidly" …. how rapidly? Please clarify.*

After the test parcel the number of iterations with the refined entrainment formulations is set to 2 because more iterations have no impact. This is now mentioned in the text. Also a flow diagram is added (suggested by reviewer 2) to clarify the role of the different updrafts and iteration procedure.

*- Section 2.2.1, eq. 9: Please state the z-range of validity (e.g. is it valid for z <=z_i,dry , then zero elsewhere?)*

Done. We now also mention that z_i,dry is the height where the dry updraft stops rising to explain that there is no need for a dry updraft entrainment above this height

*- Section 2.3, line 305: lineair -> linear*

Done

*- Section 2.4, eq. 32: Should it be l_h instead of l_{m,h}? Since this equation refers to theta_l (not momentum)?*

You are right! Changed to l_h

*- Section 3.1, line 474: I suggest to replace "cherry picking" by a less informal expression.*

Done. We now use: to avoid a possible focus on the best results

*- Fig. 12: Please explain the black dots.*

Good point. We now mention in the caption that these dots are the mean of the modelled dimensionless gradients

*- Fig. 19: If possible, I suggest that a vertical scale be added to the vertical axis of the figures in the 3 rd and 4th row (the vertical cross sections).*

Done. This indeed helps to understand the plot more easily.

*- Line 653: I'm not sure I understand "blocky pattern", so I suggest a clarification, or instead a less informal expression*

Replaced by noisy.

Reply to reviewer 2 of gmd-2021-184

We thank reviewer 2 for the positive words and the useful comments and recommendations which helped to substantially improve the paper. Below we describe how we addressed your general comments and specific points. Hopefully, you can now accept our paper for publication.

Kind regards

Wim de Rooy and co-authors

Reply to the general comments.

As described below in more detail in the reply to the specific points, we clarified and extended the description and the underlying assumptions of the various updraughts. Most importantly: We adopted your good suggestion to provide a flow diagram of the subsequent steps in the shallow convection scheme.

We now also mention the momentum mixing by the convection scheme, although no specific evaluation is presented in this study. The convection scheme in both cy40REF and cy40NEW mixes momentum in a similar way as temperature and humidity although we know this will be a clear overestimation, as momentum mixing by convection is less efficient. This is now also mentioned in the text (section 2.1). Currently we are studying, the indeed important subject, how to incorporate convective momentum mixing in a more realistic way.

Below follows our response to your specific comments.

*1. line 43, "cloud, turbulence and cloud scheme": two cloud schemes, one of which should be convection!*

Done

*2. lines 138 and 202: good that you ackowlegde this inherent dependency on the height of the lowest model level in the entrainment rate but the dependence of the initial parcel properties feels like a much stronger one that you don't seem to worry about. Building in an unnecessary resolution dependence always seems like a bad idea. A physical height, such as the top of the surface layer would seem much better, given you are adding perturbations scaled by the surface fluxes which will quantify the near-surface gradients well enough*

It seems plausible that a lower resolution, that is a higher first model level, results in a larger excess value of the updraft (compared to the environment). This because of the decrease of grid box temperature and humidity values with height. However, due to the $z^{-1}$ dependence of the entrainment formulation, the excess values decreases with a similar amount as the background. This is explained in detail in Appendix A of Siebesma et al. JAS, 2007 (A Combined Eddy-Diffusivity Mass-Flux Approach for the Convective Boundary Layer). As a result the initialization is rather independent on vertical resolution. This is now also mentioned in the text.

*3. line 141: are the updraught area fractions really constant with height or does Table 1 show the initial fractions? Later on you are explicit that this is the case for the dry updraught (line 236) but for the moist*

*updraught, given you have separate w and M profiles at least in the cloud layer, then that would imply you have an updraught area that varies? I can't see any use made of it, though, such as in the cloud scheme (see additional comment below)?*

In hindsight we fully understand that the use of the area fraction is confusing. We added text to clarify this. The area fraction of the dry updraft is used for the initialization of temperature and humidity at the lowest model level. In addition, the dry updraught area fraction is applied at all (relevant) heights to determine, together with the vertical velocity, the mass flux. The area fraction of the moist updraft is just used for the initialization of the lowest model level. Mass flux increases linearly to the closure value at cloud base. In the cloud layer, we follow the approach of de Rooy & Siebesma 2008, i.e. the dimensionless mass flux profile is depending on a bulk chi-critical value. You are right that with the mass flux and vertical velocity, a changing updraft area fraction with height could be deduced. But apart from the initialization, this area fraction is never used. The updraft area fraction in a mass flux convection scheme can be seen as a rather artificial parameter, with the primary use of getting the right turbulent transport. Comparable to the situation in LES, where different sampling techniques (with different area fractions) can be used to estimate organized turbulent transport. A larger area fraction combined with a smaller excess will result in a similar turbulent transport as a smaller area fraction with a larger excess. Ultimately, the total turbulent transport which determines via its vertical divergence the tendencies of the prognostic variables, is the most important. An example of the problematic use of the updraft fraction for the cloud fraction is stratocumulus, where the area fraction would be 1 in the cloud layer. Area fraction of 1 would suggest resolved convection.

*4. line 143: I don't understand the need for an a priori diagnosis of regime. Why not initialise both dry and moist plumes at the surface, calculate their evolution with height and from that diagnose if clouds are possible (based on the moist parcel reaching its LCL)? Is it just to save cpu time?*

Indeed, in principle there is no need to first diagnose the regime. However, to determine the ultimate entrainment formulations we first need an estimate of the inversion height. This estimate is provided by the test parcel which uses a formulation that that is not dependent on the inversion height. Because the test parcel is initialized with a relatively large excess and has a relatively small entrainment value we can be sure that if the test parcel does not experience condensation, neither will the parcel with the inversion height dependent entrainment. So indeed this is just to save computational time. This is now added in the text (and see also the new flow diagram, discussed below).

*5. section 2.1.1: I'm confused by this description of the different parcels. Perhaps a logic flow diagram would help but several questions arise. (i) you say the test parcel is used to determine an estimate of the inversion height, so how can the moist updraught LCL come into this (line 179)? (ii) in line 190 you say "this iteration process converges very rapidly" but don't say anything about what steps are iterated and how the iteration is monitored or convergence measured. (iii) can you really be confident that if the test updraught doesn't reach its LCL then the moist parcel will not reach its own? Given the different formulations this doesn't seem certain. (iv) around line 200 you suggest the dry parcel cannot reach zi,dry+a2 but isn't zi,dry just an estimate from the rather different test parcel (as you can't know the inversion height for the dry parcel without knowing dry), so what stops the dry parcel below zi,dry + a2 in practise? (v) in (10) is zlcl the LCL of the test parcel as described in line 183? If so it would be good to make that definition of zlcl explicit. But I'm concerned that you specify a change in the moist updraught's entrainment rate at zlcl even though its LCL is likely to be different, not least because of the different*

*fractional areas and sub-cloud entrainment rates. Does this not matter that the moist parcel's LCL may not coincide with where you change its entrainment rate?*

We fully agree that this section was unclear and adopt your idea to present a flow diagram. We just mention here that the inversion height of the test parcel is only used as a first estimate. Afterwards, both the dry and moist inversion height are refined separately during the iteration process (2 steps). Also here text is added.

*6. line 216:, "deeper boundary layers will contain larger updrafts with relatively small entrainment values": the references supplied don't actually show that this dependence is wrong - perhaps the specific formulation in the REF scheme is not good for the ARM case, or do you have other concerns?*

Although the correlation is not very strong we indeed have seen a hint in LES of a dependence of the entrainment at cloud base on the sub-cloud boundary layer height. However, in the references we provide only LES profiles of entrainment rate, revealing that 0.002 is a reasonable value for cloud base. The references contain no plots concerning the variation of cloud base entrainment rate with sub-cloud layer height. Or have we overlooked a particular reference?

The particular choice to return to a more simple constant value at cloud base is indeed a pragmatic one, based on the ARM case. More investigation is needed to establish an adequate description of the entrainment at cloud base and this is now also mentioned in Appendix B.

*7. line 217, "Eq. (10) shows an inverse correlation between updraft vertical velocity and entrainment magnitude": it doesn't actually, but I think you are motivating its shape from expected vertical velocities?*

Indeed, we meant that the shape of the entrainment formulation reflects the expected vertical velocity profile. Now changed accordingly in the text.

*8. line 269, "from there mass flux decreases linearly to 0 at cloud layer top": this sounds like a rather crude assumption and no massflux profiles are shown to illustrate its success or otherwise. For example, I might expect the massflux decrease to be constrained much closer to a sharper inversion, eg one maintained by radiative cooling of a stratiform layer at the cumulus top (as in the transition SCM cases shown in 3.3) than in the ARM case where shallow cumulus detrainment into a stable stratification is probably what determines the inversion thickness*

We agree that this linear decrease is a simplification, as also stated and illustrated in de Rooy & Siebesma 2008 (this paper also shows a comparison of LES and simplified non dimensionless mass flux profiles including the linear decrease for the ARM case). On the other hand, a large part of the total shape is determined by the increase or decrease half way the cloud layer. The applied dependence of this mass flux in the middle of the cloud layer shows a very strong correlation with LES for a wide variety of shallow cumulus case (see e.g. Boing et al. 2012). Moreover, if we look at observations of the trade wind cumili mass flux (Lamer et al. 2015), we see that the vast majority of the observations can be captured well with such a simplified mass flux profile (now added in the text).

In cases of stratocumulus, chi_crit is high and consequently there is no decrease of mass flux halfway the cloud layer. For stratocumulus cases we should rely on other LES sampling methods than applied in this paper. Also, for rather shallow stratocumulus layers, as studied in this paper, the vertical discretization in the cloud layer with only a few layers does not really allow a complex description of the

mass flux profile in the cloud layer and the assumption of the linear decrease in the upper part of the cloud layer seems justifiable. Finally, and more pragmatically; with the current set up, the simulation of such cloud layers, including the top entrainment, seems to be reasonable.

*9. line 357, "presume a Gaussian PDF": this seems like a big assumption, especially for cumulus clouds where the pdf is quite likely to be skewed?*

You are right that in reality the PDF will be (very) skewed in cumulus convection. However, this skewness becomes important when the cloud fraction is low (now mentioned in the text) and the impact of these very low fractions on model results will mostly be modest (see e.g. Bougeault 1981, Modeling the trade wind cumulus cloud layer part 1). Nevertheless, we will investigate the use of a skewed PDF in a future study and now also mention it as a possible solution for the missing (very) low cloud fractions in the upper part of the cloud layer in the ARM case.

*10. Fig 4: I think you could usefully and safely (ie without cluttering the plot) add the dry and moist massflux scheme components (eg as dotted and dashed lines) for the cy40 simulations, so we can see the relative contributions. I think that would give valuable insight into the workings of the overall parametrization*

Done. Indeed this information is valuable as also mentioned in the reply of your question 12.

*11. line 460: you give the horizontal resolution but not vertical. Including the height of the bottom grid-level is also clearly important for the parcel initial properties.*

We now provide the vertical number of levels and the height of the lowest model level for all cases.

*12. line 494, "inclusion of the energy cascade": this does clearly improve the ultimate fluxes but it would be good to know it also improved the TKE profiles, which are not currently shown, and so improving performance for the right reason. For example, you also speculate (line 517) that "a plausible explanation for the presence of diffusive transport...are (dry) updrafts terminating around the inversion height" but could you not be underestimating the transport by those dry updraughts themselves (again, showing the break down in Fig 4 would be instructive here)?*

This is also a good point. Unfortunately, it is not clear how the TKE that the NWP model should produce, could be diagnosed from LES. In LES, TKE is partially parameterized and partially resolved. We do not see a straightforward way to estimate the LES TKE (or some LES small scale diffusive transport) around the cloud base that could be compared to the NWP model. However, to show at least the impact of the energy cascade on the eddy diffusivity turbulent transport in the Harmonie-Arome model, we added a plot which reveals the increased eddy diffusive transport especially around the inversion. As we now provide the turbulent transport by the dry updrafts in Figure 4, like you suggested, we also mention that the strong decrease of the dry updraft indeed corresponds with the layer with substantial diffusive transport in LES, despite the strong inversion.

*13. line 550, "underestimation of low values of cloud fraction in the upper part of the cloud layer": it is hard to work out the contour interval from the colourbar but the LES looks only to have a cloud fraction of a few percent, which could be similar to your moist updraught area. Are you not missing this (highly skewed, see above) contribution to cloudiness?*

Yes you are right. The convection scheme is active in a much deeper layer than visible from the cloud fraction plot, but still we do not see any substantial cloud fraction in the upper part. The updraught area itself is not calculated explicitly (although it could be deduced) nor used directly, e.g. with some kind of pre-factor, for the calculation of the cloud fraction. In the variance plot of Fig. 10a, it becomes clear that the variance itself is largely underestimated in the area where we lack the low cloud fraction. In the Conclusions section (line 668) we mention a plausible improvement; the increase of the eddy turn-over time which seems to be on the low side. Another possible improvement we mention is to increase the local maximum around cloud top for the energy cascade function. The last options we now mention, as you suggest, is to use a skewed PDF. Note that although there is a clear mismatch with LES, the low cloud fractions will only have a limited impact on other aspects of the model due to their small influence on radiation.

*14. line 579: please could you give more detail of how the SCM is forced from RACMO (horizontal advection, surface fluxes or interactive land surface?), when are the forecasts initialised and how long are they for?*

The SCM uses the advection from the RACMO model. The surface is initialized from the RACMO model but the surface fluxes are calculated by the 1D model itself. Forecasts 72 hrs ahead are made every day at 12UTC. This is now all mentioned in the text. Note that the comparison with theoretical flux gradient relationships involves no observations. Therefore, the exact forcing of the SCM by the host model is less critical.

*15. line 601: as noted in Beare and MacVean for GABLS, many NWP centres find they have to bias their turbulence scheme away from LES in order to improve objective verification of the forecasts. Is this not the case for you and, if not, do you have any insights as to why?!*

I assume you mean that reduced mixing might lead to a better description of the stable profiles but also decreases the filling of lows and therefore results in a more active, less smooth, model behavior. As a consequence of the less smooth fields, objective model verification reveals a higher standard deviation (double penalty) of output parameters. Indeed this is not something we experience. Part of the explanation might be that the effect of the turbulence modifications cannot be summarized by simply less mixing. As shown in Fig. 12 by the dimensionless gradients, the new set-up actually result in more mixing in the surface layer. On the other hand higher up in the atmosphere, e.g. near the inversion, mixing is decreased. Something that supports the latter is the decrease of the limit length scale for the free atmosphere in Eq. (16) which has no impact near the surface.

*16. line 623: I don't see why you would invoke a different set-up in this case only, especially for the REF scheme? Isn't the purpose to illustrate operational performance in simple test cases? It is then not clear, when you say (line 628) "Based on the considerations above, the stratocumulus regime with only a wet updraft is removed in cy40NEW", whether that is just in the SCM simulation or is this the motivation for this change in all tests?*

This is indeed not stated clear enough. The removal of the regime with just the moist updraft is part of the modifications in cy40NEW and therefore applies to all results with cy40NEW in the paper. This is now mentioned explicitly in the text.

*17. line 636, "Key aspect of the large improvement with cy40NEW is again the better preservation of inversion strengths": this statement would be much stronger if backed up by a sample profile or cross section showing this sharper inversion.*

We added a plot referring to the case in Fig 21 which confirms the increased inversion strength.

*18. line 638, "removal of the HARATU updates": these were reported (line 278) as being needed to alleviate problems with wind speeds so how do those look in cy40NEW? You do (finally, line 698) say the performance is maintained but in what metric (diurnal cycle of wind speed bias would be a good one to show)?*

Indeed, near-surface variables, like 10m wind speed, are influenced by the model update. We now describe the main (small) overall impact on 2m temperature and humidity as well as 10m wind speed. In general the wind speed itself increases somewhat due to the modifications but (amplitude of) the diurnal cycle is comparable. Small improvements as well as small deteriorations in 10m wind speed can be found depending on the choice of the month or domain. Therefore, we think that adding e.g. a diurnal cycle of the wind speed bias plot would require quite some additional explanation and would distract from the main subject. We now also mention that near-surface parameters are strongly influenced by surface processes (not involved in this study) and representation mismatch between grid box and observation site conditions.

*19. Table D1, Shallow convection scheme, cy40REF: I suspect the formula for lcl has too many layers of subscript in zzlcl?*

Yes you are right. Solved.

---

## Author Response (AR2)

Dear Sylwester Arabas,

Thank you for carefully reading our previous version of the paper. Below we describe how we addressed your comments. Please let us know if something is still missing or should be corrected.
Hopefully, the paper is now ready for publication.
Kind regards,
Wim

*Dear Authors, Thank you for the revision and congratulations for receiving "accept as is" recommendation from the reviewer.*
*Let me ask for addressing the following technical issues before we proceed:*

*- is the space between "van" and "'t" intentional in "Bram van 't Veen" author's name? (it is not there on LinkedIn profile, for example: https://nl.linkedin.com/in/bram-van-t-veen-21a30816a)*
I checked it with him and there should be a space between "van" and "'t"
*- please provide figures 3 and 20 in vector format*
Figure 3 is now provided in vector format. Figure 20 was already in vector format (eps), maybe you meant Fig 14, which was in png. This plot is now included as pdf. Hopefully, the quality is sufficient.
*- suggest increasing font size in the legend of Fig 8*
Done
*- spell Python with capital P (Acknowledgements)*
Done
*- please add DOI identifiers for all journal entries in the references*
Done
*- please unify journal acronyms in references, journal names should be abbreviated according to the Caltech database: http://library.caltech.edu/reference/abbreviations/*
Done
*- please unify capitalization in references (All Words In Titles vs. Only first word)*
Done (all words)
*- for web resources cited in the references, please add "Access date" field (see https://www.geoscientific-model-development.net/submission.html#references)*
Done
*- for web entries such as Mallardel & Tudor 2004, it would be of great value to ask the authors to deposit their report on arXiv or similar permanent repository to allow for proper citation*
A good point but considering the (response) time, this is difficult to realize. All current web entries have a permanent character though. Hopefully, you can accept the entries as they are now.
*- please double check spelling, capitalization and punctuation in references (e.g., "and andL. Papritz", kNMI, kVR, "convection,similarity")*

Done
*- for Tompkins 2005 report, please use the persistent URL: https://www.ecmwf.int/node/16958*
Done
*- when citing references within parenthesis, please eliminate repeated closing parentheses "))" - over 40 occurrences in the paper, for solution see examples of how to use \citep{} in "LITERATURE CITATIONS" block in the template.tex file of the Copernicus LaTeX package (https://www.geoscientific-model-development.net/Copernicus_LaTeX_Package.zip)*
Done
*- when citing references without parentheses, please use \citet{} instead (e.g., "similar to (Soares et al., 2004)" -> "similar to Soares et al. (2004)")*
Done
*- in Table D1, please rephrase statements in the right column to make all start with a noun (e.g., "reduce" -> "reduction"; "now consistency" -> "consistency"; "improves" -> "improvement", ...)*

Done